# BYOL: Bring Your Own Language Into LLMs

## Abstract

Large Language Models (LLMs) exhibit strong multilingual capabilities, yet remain fundamentally constrained by the severe imbalance in global language resources. While over 7,000 languages are spoken worldwide, only a small subset (<100) has sufficient digital presence to meaningfully influence modern LLM training. This disparity leads to systematic underperformance, cultural misalignment, and diminished accessibility for speakers of low-resource and extreme-low-resource languages. To address this gap, we introduce **B**ring **Y**our **O**wn **L**anguage (BYOL), a unified framework that enables scalable, language-aware LLM development tailored to each language's digital footprint. BYOL begins with a language resource classification—mapping languages into four tiers (Extreme-Low, Low, Mid, High) based on curated web-scale corpora, and uses this classification to determine the appropriate integration strategy. For low-resource languages, we propose a full-stack data refinement and expansion pipeline, combining corpus cleaning, synthetic text generation, continual pretraining, and supervised finetuning. Applied to `Chichewa` and `Māori`, this pipeline yields two language-specific LLMs that achieve ∼12% average improvement over strong multilingual baselines across 12 benchmarks, while preserving English and multilingual capabilities via weight-space model merging. For extreme-low-resource languages, we introduce a translation-mediated inclusion pathway, demonstrating with `Inuktitut` that a tailored MT system can deliver +4 BLEU improvement over a commercial baseline, enabling high-accuracy LLM access in settings where direct modeling is otherwise infeasible. Our results show that BYOL offers a practical, extensible, and data-efficient recipe for expanding LLM capabilities to the long tail of the world's languages. We will release human-translated versions of the Global MMLU-Lite benchmark in `Chichewa`, `Māori`, and `Inuktitut`, and make our codebase and models publicly available.

## 1 Introduction

LLMs have achieved remarkable gains across natural language processing tasks, driven by large-scale pretraining on multilingual web corpora (Gao et al., 2020; Weber et al., 2024; Abadji et al., 2022; Soboleva et al., 2023; Penedo et al., 2024b; Soldaini et al., 2024; Penedo et al., 2024a). However, they are strongly affected by the uneven distribution of digital text across languages (Joshi et al., 2020; Penedo et al., 2025). While over 7,000 languages are spoken worldwide[1], only a small fraction dominates the web (e.g., ∼90% of Common Crawl text comes from just twenty languages[2]). As generative AI increasingly becomes a general-purpose technology, this imbalance makes access to its benefits language-contingent, thereby reinforcing a systematic divide between high-resource languages (with abundant digital text) and low-resource languages (with minimal digital presence) (coh, 2024; Peppin et al., 2025).

Crucially, this gap is not merely a quality issue. In countries where most people operate in local languages, a primarily English-centric LLM ecosystem effectively gates the benefits of AI behind a linguistic barrier. Malawi is a concrete illustration: although English is an official language, everyday communication is dominated by local languages such as Chichewa; fewer than 4% of the population speaks English, and current LLM support for Chichewa remains limited. Without practical, repeatable ways to bring such languages

---

[1] https://www.ethnologue.com/
[2] https://commoncrawl.github.io/cc-crawl-statistics/plots/languages.html

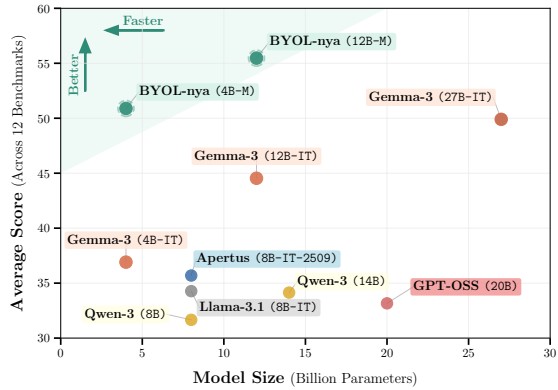 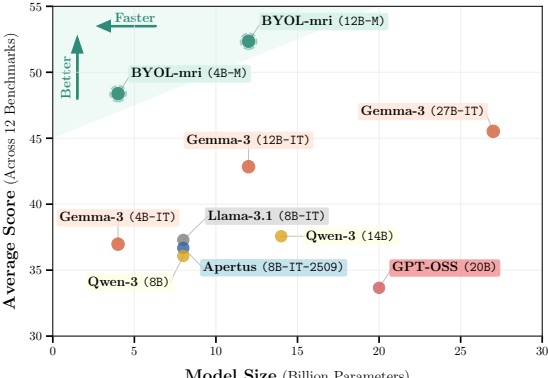

Figure 1: **Performance comparison of LLMs** on `Chichewa` (left) and `Māori` (right). On both languages, our `BYOL` models deliver strong results; notably, the 4B variants outperform the ∼7× larger Gemma-3 (27B-IT).

into LLMs, the resulting imbalance becomes a widening AI diffusion divide (Misra et al., 2025a;b), with downstream implications for access to AI-enabled services in education, health, legal systems, and economic productivity (Appel et al., 2025; Patwardhan et al., 2025).

The language divide also has broader technical consequences. LLMs trained primarily on English and a small set of high-resource languages show degraded performance on underrepresented languages (Fig. 1), limited grounding in local context, and amplified cultural and epistemic biases (coh, 2024; Singh et al., 2024a). The lack of clean, sufficiently sized corpora constrains multilingual generalization (Holtermann et al., 2024; Maini et al., 2025; Xuan et al., 2025; Ahuja et al., 2024). Even when some data exists, expanding models to include more languages introduces the curse of multilinguality (Chang et al., 2024; Arivazhagan et al., 2019; Conneau et al., 2020; Pfeiffer et al., 2022), where as language coverage increases, overall performance declines. Despite claims of broad multilingual support, most LLMs are evaluated primarily on high-resource languages, leaving the long tail of Low-Resource Languages (LRLs) untested (Üstün et al., 2024; Adelani et al., 2025; Singh et al., 2024a; Ahuja et al., 2024; Hernández-Cano et al., 2025).

Beyond accuracy, speakers of underrepresented languages also face practical disadvantages, including higher inference costs and latency due to inefficient tokenization (Ahia et al., 2023; Arnett & Bergen, 2025; Arnett, 2025; Abagyan et al., 2025; Hernández-Cano et al., 2025), and exclusion from safety-critical applications (Peppin et al., 2025) as models insufficiently tuned for a language may distort meaning and produce harmful content (Deng et al., 2024; Yong et al., 2023; Peppin et al., 2025). Communities in LRL speaking countries (Misra et al., 2025b) also face structural barriers, i.e., limited access to compute, data, and research ecosystems, that further widen the technological gap (OECD, 2023; Nekoto et al., 2020). Addressing this challenge requires far more than scaling existing multilingual models. We argue that the solution must be a language-centric, efficient model development approach that adapts to each language's digital footprint, resource quality, and practical constraints.

In this work, we introduce a unified framework, Bring Your Own Language (BYOL), designed to systematically enable LLM capabilities for low-resource and extreme-low-resource languages. **First**, we propose a **language resource classification** framework that maps each language to one of four tiers (Extreme-Low, Low, Mid, High) based on its effective digital footprint in curated web-scale corpora. This classification guides integration strategies: direct *finetuning* for mid/high-resource languages, additional *continual pretraining* for low-resource languages, and *translation-based inclusion* for extreme-low-resource cases. **Second**, for languages with limited noisy but usable corpora (low-resource tier), we develop a **data refinement and expansion pipeline** that cleans, augments, and enhances native-language text to support continual pretraining and downstream finetuning. We demonstrate this pipeline through two full-stack case studies[3]: a `Chichewa LLM` (named

---

[3]Chichewa (ISO 639-3: `nya`) is a low-resource Bantu language of Malawi, and Māori (ISO 639-3: `mri`) represents a revitalized Indigenous language of New Zealand. Chichewa and Māori were selected as representative low-resource languages from distinct linguistic families, allowing evaluation across typologically diverse, underserved languages.

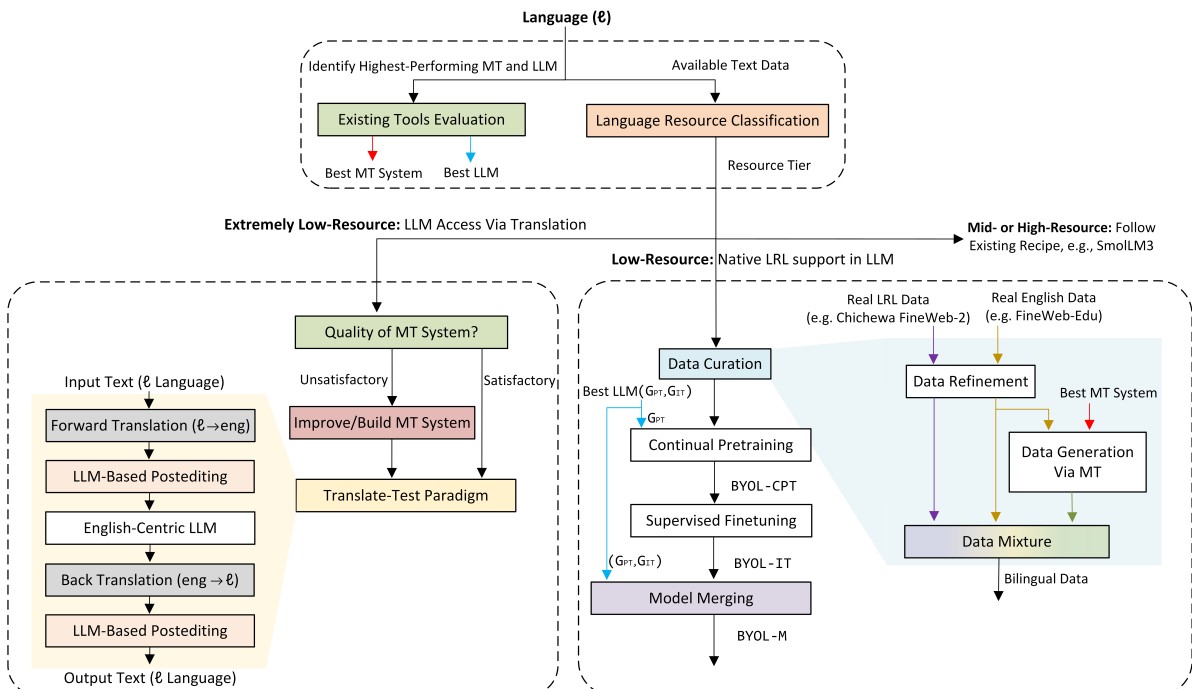

Figure 2: **Overview of the BYOL pipeline.** The system classifies a target language $\ell$ by resource tier and selects the appropriate adaptation pathway. $G_{\mathrm{PT}}$ and $G_{\mathrm{IT}}$ denote the base and instruction-tuned variants of the generalist LLM selected through the initial tool evaluation.

`BYOL-nya`) and a `Māori LLM` (`BYOL-mri`), each achieving roughly 12% average improvement over strong multilingual baselines across 12 benchmarks (Fig. 1). **Third**, for languages with negligible digital presence (*extreme-low-resource* tier), we introduce a **translation-mediated inclusion pathway** that enables access to LLM capabilities via high-quality forward- and back-translation. Using `Inuktitut`[4] as a case study, we train a translation system achieving a ∼4 BLEU improvement over a commercial baseline and show that translation-mediated LLM use yields a ∼14% accuracy gain over direct inference. **Finally**, to support open, comparable evaluation for future research, we release *human-translated versions* of Global MMLU-Lite (Singh et al., 2024a) in `Chichewa`, `Māori`, and `Inuktitut` languages. Our overarching goal is to demonstrate a scalable and extensible recipe for supporting the world's LRLs that, in contrast to generic multilingual scaling, shows the promise of language-aware, resource-adaptive LLM development for *all* languages.

## 2 Bring Your Own Language (`BYOL`) Framework

**Overall pipeline.** Given a target language $\ell$, our pipeline (Fig. 2) begins by evaluating existing tools to identify the best-performing LLM and machine translation (MT) systems. Concurrently, a language resource classification module analyzes the amount of text data available for language $\ell$ and assigns it to one of four tiers: extreme-low-resource, low-resource, mid-resource, or high-resource. This classification determines the route for language adaptation. For **extreme-low-resource languages**, where textual data is insufficient for direct adaptation, access to LLM capabilities is enabled through a translation interface that follows the Translate-Test paradigm (Artetxe et al., 2023). For **low-resource languages**, which have limited but usable data, the framework employs a data-centric strategy to enable native LLM support. Languages classified as **mid- or high-resource**, which are typically well represented in multilingual models (Abdin et al., 2024; Agarwal et al., 2025; Team et al., 2025; Yang et al., 2025; Liu et al., 2024; Grattafiori et al., 2024), fall outside the scope of this work.

---

[4]Inuktitut (ISO 639-3 code: `iku`) is an Indigenous Inuit language spoken in Inuit Nunangat, Canada.

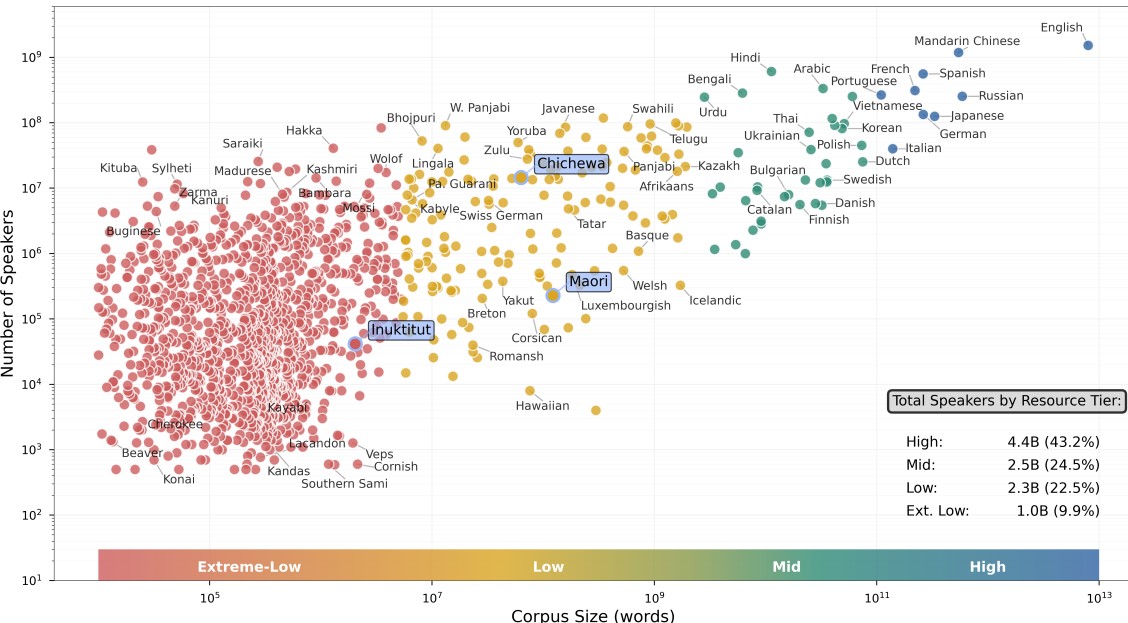

Figure 3: **Language resource classification** derived from FineWeb2 (Penedo et al., 2025). The continuous color bar shows the full spectrum of text availability, while the four discrete resource tiers (Extreme-Low, Low, Mid, High) use interpretable corpus-size boundaries to route languages through the appropriate pathway in the `BYOL` framework.

## 2.1 Initial Assessment

The assessment stage has two key modules: (1) Language Resource Classification, which assigns $\ell$ to a resource tier, and (2) Existing Tools Evaluation, which benchmarks the performance of existing MT and LLM systems for $\ell$.

### 2.1.1 Language Resource Classification

The taxonomy of Joshi et al. (2020) categorizes languages into six resource levels and remains a foundational reference for language classification. However, it predates the modern LLM era, and the web-scale data available for many languages has changed substantially since its publication. We therefore revisit language resource classification using FineWeb2 (Penedo et al., 2025), a curated multilingual corpus derived from and deduplicated across Common Crawl (Common Crawl). FineWeb2 provides a consistent estimate of the effective digital footprint for more than 1,000 written languages. For each language, we compute the total word count as a proxy for corpus size and pair it with speaker population to provide a two-dimensional view of digital representation (Fig. 3).

We group languages into four corpus-size tiers using interpretable boundaries that reflect the distribution in Fig. 3 and align with practical distinctions observed in multilingual LLM pretraining (Kudugunta et al., 2023; Penedo et al., 2025).

- **Extreme-Low-Resource** ($\leq 5 \times 10^6$ words): Languages with negligible digital presence and minimal-to-no LLM exposure. For these languages, native integration into LLMs is currently infeasible, and MT-based access is the most practical route.

- **Low-Resource** ($5 \times 10^6 - 2 \times 10^9$ words): Languages with limited but usable textual data, making them candidates for native inclusion in LLMs through targeted continual pretraining.

- **Mid-Resource** ($2 \times 10^9 - 10^{11}$ words): Languages with substantial textual resources and moderate-to-strong LLM coverage, for which light adaptation (e.g., domain-specific finetuning) can typically close most of the remaining task-specific performance gap.

- **High-Resource** ($> 10^{11}$ words): Languages with abundant, high-quality web-scale corpora that enjoy comprehensive LLM support across diverse tasks.

These tier boundaries are indicative rather than absolute, since language resources evolve continuously as new text becomes available. In practice, the effort required for a language to approach reference English performance depends not only on corpus size, but also on linguistic characteristics, writing system, and the extent to which it can benefit from cross-lingual transfer. The continuous color bar in Fig. 3 reflects this spectrum of digital scarcity, while the four tiers introduced in this paper provide a simple, actionable routing scheme that guides the integration choices of the `BYOL` framework. Figure 3 also shows that speaker population and corpus size are not strictly correlated: some languages with relatively small populations (e.g., Icelandic) have strong web presence, whereas others with millions of speakers (e.g., Saraiki, Kituba) remain digitally underrepresented.

### 2.1.2   Existing Tools Evaluation: LLMs and MT Systems

The second module identifies which LLMs and MT systems currently support language $\ell$ and measures their performance. This evaluation serves three purposes. (1) For low-resource languages, MT systems can be used to generate synthetic training data by translating high-quality corpora from pivot languages (e.g., English or Spanish) into $\ell$, supplementing limited real text. (2) It identifies the highest-performing LLM with *any* measurable representation of language $\ell$, providing a baseline model for subsequent adaptation. (3) For extreme-low-resource languages lacking direct LLM support, MT systems serve as a fallback interface, enabling indirect access to LLM capabilities via translation.

A major challenge is the lack of benchmarking datasets, which affects thousands of languages, as existing resources (e.g., FLORES-200 (NLLB Team et al., 2024)) cover only a small subset. Creating new parallel benchmarks for each language is prohibitively costly and time-intensive. We therefore adopt a scalable, reference-free evaluation framework.

**Round-trip translation as proxy evaluation.** We employ the round-trip translation (RTT) strategy (Zhou et al., 2023b). A sentence from a pivot language (e.g., English[5]) is translated into $\ell$ and then back into the pivot language. The reconstructed sentence is compared with the original source to measure how well meaning is preserved through the round-trip translation cycle. RTT is domain-agnostic in its original form. However, translation quality varies significantly across domains (Koretaka et al., 2023; Saunders & DeNeefe, 2024; Shen et al., 2021; Saunders, 2022). To address this, we extend RTT with a domain-conditioned evaluation that enables fine-grained analysis of how translation models generalize across different domains. Overall, the process is defined as:

$$\text{RTTScore} = \frac{1}{|D|} \sum_{d \in D} \frac{1}{N_d} \sum_{i=1}^{N_d} \mathcal{M}\left(s_i^{(d)}, \mathcal{T}_{\ell \to \text{eng}}\left(\mathcal{T}_{\text{eng} \to \ell}\left(s_i^{(d)}\right)\right)\right), \tag{1}$$

where $D$ is the set of domains, $s_i^{(d)}$ is the $i$-th sentence sampled from domain $d$, $N_d$ is the number of sentences in that domain, $\mathcal{T}_{(\cdot)}$ denotes an MT engine or an LLM, and $\mathcal{M}$ represents a fidelity metric such as SacreBLEU (Post, 2018), chrF++ (Popović, 2017), or embedding-based cosine similarity. To operationalize this framework, we introduce a dedicated benchmark for cross-domain RTT evaluation.

**RTTBench-Mono: domain-balanced monolingual dataset.** We develop RTTBench-Mono, a 1,250-sentence English dataset across 25 domains from NVIDIA's taxonomy[6] after excluding the adult category. For each domain, 50 sentences are generated using `GPT-4.1` with varying lengths and syntactic complexity. The dataset provides balanced coverage across diverse topics and serves as a standardized source for domain-conditioned RTT evaluation. We validate dataset quality using classifier-based and embedding-based checks; details on validation methodology, prompt design, and the procedure to prevent domain drift are provided in Appendix E.1.

---

[5]While English is used as an example pivot, any high-resource language that is typologically and culturally closer to the target language can be substituted.

[6]https://huggingface.co/nvidia/domain-classifier

## 2.2 Low-Resource Pathway: Native Language Support in LLMs

Training an LLM from scratch for a low-resource language is infeasible due to limited text. We therefore begin with a multilingual *generalist* LLM that shows preliminary knowledge of $\ell$, identified through the tool-assessment procedure in Sec. 2.1.2. We transform this model into a *language-specific expert* through three stages: continual pretraining, supervised finetuning, and model merging.

**Continual pretraining (CPT).** FineWeb2 (Penedo et al., 2025) is the primary source of multilingual non-English text for pretraining, but its coverage varies substantially across LRLs (Fig. 3). To expand the corpus, we translate the English FineWeb-Edu dataset (Penedo et al., 2024a) into the target $\ell$ using the best-performing MT system identified in Sec. 2.1.2. This resulting synthetic text is mixed with the real LRL data. We also include a random subset of English FineWeb-Edu to preserve the baseline LLM's English competence and cross-lingual capabilities in the final model. Although FineWeb2 applies rule-based filtering, its text still contains formatting errors, noise and limited coherence. We therefore refine all the pretraining data through guided rephrasing using a large multilingual LLM. Each text sample is treated as an initial draft and rewritten to improve clarity and structure, while expanding on-topic content. Additionally, toxic and harmful material is removed. The final CPT corpus consists of: (1) refined real LRL text from FineWeb2, (2) synthetic LRL data translated from the refined FineWeb-Edu corpus, and (3) refined real English data from FineWeb-Edu (Penedo et al., 2024a). Further details on CPT dataset construction are provided in Appendix B.1, and the data-refinement prompt is given in Appendix F.1. We perform CPT separately for each target language using its corresponding bilingual data mixture. This stage strengthens the model's internal representation of $\ell$ and mitigates language drift, preventing the model from unintentionally switching to other languages mid-sentence during text generation.

**Supervised finetuning (SFT).** We assemble a bilingual instruction dataset for SFT. When available for the target language, we source native-language $\ell$ samples from the Aya dataset (Singh et al., 2024b) and Smol (Caswell et al., 2025). Because most LRLs lack sufficient instruction data, we translate instruction-response pairs from five high-resource Aya languages, as well as a subset of SmolTalk2 (Bakouch et al., 2025), into $\ell$. We also include a portion of English instruction data from SmolTalk2 to maintain cross-lingual alignment. Dataset composition is summarized in Appendix B.2. Each CPT model is finetuned on this data mixture to enhance its instruction-following and response-generation capabilities in the target language $\ell$.

**Model merging.** Model merging combines a multilingual generalist model with a language-specific expert directly in weight space. This yields a unified network that gains native proficiency in $\ell$ while retaining the multilingual behavior of the generalist model. Let $G_{\mathrm{PT}}$ and $G_{\mathrm{IT}}$ denote the pretrained and instruction-tuned variants of the multilingual generalist model $G$. Let $E_\ell$ represent the language expert for $\ell$, obtained by applying CPT followed by SFT to $G_{\mathrm{PT}}$. Because all models share the same initialization, their parameter spaces remain aligned, allowing linear combination in weight space. We define the merged model as:

$$M(\alpha, \beta) = G_{\mathrm{PT}} + \alpha\left(G_{\mathrm{IT}} - G_{\mathrm{PT}}\right) + \beta\left(E_\ell - G_{\mathrm{PT}}\right), \tag{2}$$

where $\alpha$ and $\beta$ are positive scaling coefficients. The first term $(G_{\mathrm{IT}} - G_{\mathrm{PT}})$ transfers the instruction-following behavior of the generalist model $G$, while the second term $(E_\ell - G_{\mathrm{PT}})$ injects the language-specific knowledge. This approach brings low-resource language expertise into the baseline LLM while preserving its multilingual and safety behaviors, without requiring additional training or alignment steps (Huang et al., 2024).

## 2.3 Extreme-Low-Resource Pathway: LLM Access via Machine Translation

For extreme-low-resource languages, available text is insufficient for direct model adaptation (Fig. 3). LLMs trained on such limited data produce unintelligible outputs if used directly. Therefore, we adopt the *Translate-Test* paradigm (Artetxe et al., 2023), where input text is translated into English, processed by an English-centric LLM, and translated back into the source language. This enables access to advanced LLM capabilities even for languages with minimal digital presence. The effectiveness of the Translate-Test approach depends on MT quality: a usable MT system must exist for the target language, and its translations

should be sufficiently accurate, since any errors or information loss propagate to the English LLM, degrading final responses. As a case study, we develop an MT system for *Inuktitut*, and integrate it with an LLM.

**Sentence alignment in extreme-low-resource languages.** Sentence alignment identifies matching sentence pairs across bilingual documents and is critical for MT, as translation models are highly sensitive to noise and misaligned training examples. Existing approaches rely on length-based heuristics (Gale & Church, 1993), lexical overlap (Moore, 2002), or pretrained embedding models (Thompson & Koehn, 2019; Feng et al., 2022; Heffernan et al., 2022) that are typically unavailable or unreliable for extreme-low-resource languages. Therefore, we employ a multilingual LLM to perform sentence alignment directly. Even with weak understanding of a given language, an LLM can still leverage structural and contextual cues to detect cross-lingual correspondences. In our pipeline, we use `GPT-5` to extract aligned pairs from bilingual sources such as news articles and children's books. The full alignment prompt is provided in Appendix F.2.

**Synthetic data generation via back-translation.** High-quality parallel data is essential for MT training (Costa-Jussà et al., 2022; Kudugunta et al., 2023). However, extreme-low-resource languages typically have limited and domain-constrained bitext. We therefore use back-translation (Sennrich et al., 2016a; Edunov et al., 2018) to expand the available training data. We first train an initial $\ell$-*to-English* model on the existing bitext and use it to translate monolingual $\ell$ sentences into English, producing synthetic English paired with human-written $\ell$ text. The synthetic and real pairs are mixed at a 1:1 ratio (Haque et al., 2020) to balance quality and diversity.

Since the Translate-Test paradigm requires MT systems in both directions (English-to-$\ell$ and $\ell$-to-English), we construct distinct monolingual corpora for each direction. For the English-to-$\ell$ model, we rely on monolingual text in $\ell$, whose availability varies across languages (Penedo et al., 2025). For the $\ell$-to-English model, we assemble an English corpus by sampling from diverse datasets (e.g., WikiMatrix (Schwenk et al., 2021), News Commentary (Tiedemann, 2012), Global Voices (Nguyen & Daumé III, 2019)); the complete list of datasets is in B.3 (Table 13). The English samples undergo a two-stage cleanup: (1) rule-based filtering to remove duplicates, malformed text, and out-of-range lengths, and (2) LLM-based refinement to eliminate non-English or low-quality sentences and improve clarity while preserving meaning (see prompt for LLM-based refinement in Appendix F.3). The resulting clean English corpus is then back-translated using the English-to-$\ell$ model.

**LLM-based post-editing of MT outputs.** MT systems trained on limited data often produce outputs with lexical inaccuracies or subtle word-choice errors, especially across semantically similar domains (Bapna et al., 2022; Nielsen et al., 2025). In contrast, multilingual LLMs, though performing poorly at *direct* translation in low-resource languages, possess strong general-purpose reasoning capabilities, making them well suited as post-editors for refining MT outputs (Nielsen et al., 2025). We therefore employ `GPT-5` to perform light post-editing, where the model applies minimal corrections to improve grammatical accuracy, lexical precision, and overall fluency while preserving the meaning and structure of the original NMT translation. The LLM is explicitly prompted to avoid unnecessary rephrasing and instead focus on correcting mistranslated words and minor inconsistencies to enhance translation quality. The post-editing prompt template is provided in Appendix F.4.

## 3 Experiments and Analysis

To demonstrate the effectiveness of our pipeline, we evaluate two pathways: (1) adapting the LLM directly to target low-resource languages, and (2) enabling translation-mediated LLM access for extreme-low-resource languages.

### 3.1 Direct LLM Adaptation for Chichewa and Māori

#### 3.1.1 Experimental Details

**Baseline model selection.** We choose Gemma-3 (Team et al., 2025) as our starting point due to its stronger performance on Chichewa and Māori, as we shall see in ablation experiments. We compare our adapted models against several LLMs, including Llama-3.1 (Grattafiori et al., 2024), Qwen-3 (Yang et al., 2025),

GPT-OSS (medium reasoning mode) (Agarwal et al., 2025), Apertus (Hernández-Cano et al., 2025), and GPT-4o.

**Training datasets.** Our experiments use bilingual data mixtures (English and the target language) in both continual pretraining and supervised finetuning stages. For CPT, the corpus combines real LRL text from FineWeb2 (Penedo et al., 2025), synthetic LRL text obtained by translating the English FineWeb-Edu corpus (Penedo et al., 2024a), and English text from FineWeb-Edu (Penedo et al., 2024a). These components are mixed at a 1:1:1 ratio, and all text is refined using the data curation strategy described in Sec. 2.2. The final CPT mixtures contain approximately 433M tokens for Chichewa and 745M tokens for Māori.

For SFT, we assemble a bilingual instruction dataset from several sources. We use native QA pairs from the Aya dataset (Singh et al., 2024b) when available for the target LRL. To increase coverage, we translate Aya QA pairs from five high-resource languages (English, French, Dutch, Spanish, and Italian) into the target language. We also translate SmolTalk2 (Bakouch et al., 2025) into the target LRL and include its English samples to maintain cross-lingual alignment. The detailed SFT dataset composition is shown in B.2 (Table 11).

**Benchmarking datasets and evaluation metrics.** Existing multilingual benchmarks provide little-to-no coverage for most low-resource languages, and Chichewa and Māori are present only in FLORES-200 (Costa-Jussà et al., 2022) (translation) and Belebele (Bandarkar et al., 2024) (reading comprehension). To perform comprehensive evaluation, we introduce professionally translated versions of Global MMLU-Lite (Singh et al., 2024a) for both languages. We further generate machine-translated variants of ten benchmarks: ARC-Easy/Hard (Clark et al., 2018), MGSM (Shi et al., 2022), XCOPA (Ponti et al., 2020), StoryCloze (Lin et al., 2021b), PIQA (Bisk et al., 2020), HellaSwag (Zellers et al., 2019), XNLI-2.0 (Upadhyay & Upadhya, 2023), XWinograd (Tikhonov & Ryabinin, 2021), and TruthfulQA-Multi (Lin et al., 2021a). All translated benchmarks are integrated into the `lm-evaluation-harness` framework (Gao et al., 2024). We also report English performance of competing models on HumanEval (Chen et al., 2021), BBH (Suzgun et al., 2022), GPQA-Diamond (Rein et al., 2024), and IFEval (Zhou et al., 2023a). In addition, we run pairwise comparisons using an LLM-as-a-judge setup on MultiWikiQA (Smart, 2025). Throughout the paper, we report scores for each benchmark/task using its standard evaluation metric (accuracy, BLEU, chrF++, etc.). The average score (reported as a percentage) is computed by normalizing each metric to the $[0, 1]$ range and using chrF++ for the translation task. Details on evaluation benchmarks and metrics are provided in Appendix C.1 (Table 14) for base models, and in Appendix C.2 (Table 15) for instruction-tuned models.

**Hyperparameters.** We train separate models, `BYOL-nya` and `BYOL-mri`, at three different sizes: 1B, 4B, and 12B parameters. All models are optimized with AdamW ($\beta_1 = 0.9$, $\beta_2 = 0.999$) for 4 epochs. The learning rate is set to $2 \times 10^{-5}$ and gradually reduced to $2 \times 10^{-6}$ using cosine annealing, with a linear warm-up over the first 3% of training iterations. During training, we set the maximum sequence length to 4096 tokens. We use the same hyperparameters for SFT, except that we use a lower learning rate of $1 \times 10^{-5}$ and train for 2 epochs.

### 3.1.2 Performance Evaluation

**Base model results.** Tables 1 and 2 present base model comparisons on several Chichewa (`nya`) and Māori (`mri`) benchmarks. Across both languages, our `BYOL` models provide consistent gains at similar parameter scales and often surpass significantly larger models. For example, in Table 1, the 4B `BYOL-nya` model yields a 13.52 point average improvement over the 2× larger Apertus (8B) (Hernández-Cano et al., 2025) baseline and a 1.24 gain over the 3× larger Gemma-3 (12B-PT) (Team et al., 2025) model. Similarly, Table 2 shows that our `BYOL-mri` model obtains an average score of 47.72, compared to 44.88 for the Gemma-3 (12B-PT) model, while having 3× fewer parameters. Notably, our continual-pretrained `BYOL` models achieve these gains while preserving the English performance of the Gemma-3 (PT) baselines; see D (Table 16).

**Instruction-tuned model results.** Tables 3 and 4 report performance comparisons of our `BYOL` models against several competing LLMs of varying parameter capacities on Chichewa and Māori benchmarks, respectively. These `BYOL` models are obtained via supervised finetuning followed by model merging (see

Table 1: **Base model results on Chichewa** (`nya`) language benchmarks. Our `BYOL-nya` (CPT) models yield significant gains, and notably the 4B variant surpasses Gemma-3 (12B-PT), despite being 3× smaller.

| Benchmarks | | 1B − 2B Models | | | | 4B − 8B Models | | | | | 12B+ Models | | |
|---|---|---|---|---|---|---|---|---|---|---|---|---|---|
| | | Llama-3.2 (1B) | Qwen-3 (1.7B-Base) | Gemma-3 (1B-PT) | BYOL-nya (1B-CPT) | Llama-3.1 (8B) | Apertus (8B-2509) | Qwen-3 (8B-Base) | Gemma-3 (4B-PT) | BYOL-nya (4B-CPT) | Qwen-3 (14B-Base) | Gemma-3 (12B-PT) | BYOL-nya (12B-CPT) |
| Global MMLU-Lite (Singh et al., 2024a) | | 28.25 | **32.50** | 26.75 | 23.00 | 43.25 | 41.75 | 37.50 | 50.75 | **55.25** | 45.75 | 60.75 | **64.50** |
| ARC-Easy (Clark et al., 2018) | | 28.66 | 28.83 | 29.04 | **36.66** | 28.62 | 31.02 | 29.63 | 30.22 | **48.48** | 29.00 | 39.98 | **51.14** |
| ARC-Hard (Clark et al., 2018) | | 22.44 | 21.93 | 23.72 | **27.56** | 24.06 | 27.73 | 23.98 | 27.13 | **40.61** | 25.68 | 35.41 | **42.41** |
| MGSM (Shi et al., 2022) | | 1.6 | 7.60 | 2.00 | **3.20** | 8.40 | 10.40 | 10.00 | 17.20 | **31.60** | 14.40 | 44.00 | **53.20** |
| XCOPA (Ponti et al., 2020) | | 48.60 | 52.00 | 51.80 | **61.20** | 51.20 | 53.20 | 52.20 | 57.20 | **70.00** | 51.00 | 60.80 | **71.20** |
| XStoryCloze (Lin et al., 2021b) | | 48.38 | 47.45 | 50.96 | **55.79** | 50.43 | 53.74 | 49.11 | 54.40 | **65.98** | 50.83 | 63.53 | **67.90** |
| PIQA (Bisk et al., 2020) | | 51.74 | 51.25 | 51.69 | **58.54** | 51.74 | 53.86 | 51.09 | 54.57 | **63.71** | 52.39 | 58.27 | **64.96** |
| HellaSwag (Zellers et al., 2019) | | 29.19 | 29.93 | 29.05 | **37.13** | 29.56 | 31.98 | 27.81 | 33.45 | **47.31** | 30.69 | 44.09 | **51.89** |
| XNLI 2.0 (Upadhyay & Upadhya, 2023) | | 33.79 | 33.07 | 34.19 | **37.92** | 33.81 | 35.01 | 34.73 | 37.82 | **40.32** | 33.95 | 40.98 | **45.21** |
| XWinograd (Tikhonov & Ryabinin, 2021) | | 50.59 | 49.20 | 51.34 | **63.32** | 50.59 | 56.15 | 52.09 | 54.76 | **68.34** | 51.12 | 61.39 | **70.37** |
| Belebele (Bandarkar et al., 2024) | | 27.56 | **29.22** | 28.11 | 26.00 | 29.22 | 38.78 | 32.33 | 38.22 | **45.44** | 36.56 | 59.56 | **61.00** |
| FLORES-200 (Costa-Jussà et al., 2022) (nya→eng) | BLEU | 2.81 | 1.03 | 5.02 | **14.77** | 9.40 | 18.92 | 2.95 | 17.28 | **23.87** | 6.19 | 25.59 | **27.84** |
| | chrF++ | 19.75 | 15.26 | 24.17 | **38.18** | 31.38 | 42.25 | 21.97 | 40.37 | **47.95** | 27.85 | 48.91 | **51.12** |
| FLORES-200 (Costa-Jussà et al., 2022) (eng→nya) | BLEU | 0.43 | 0.04 | 0.40 | **9.53** | 0.71 | 2.16 | 0.04 | 2.24 | **12.79** | 0.07 | 9.66 | **13.82** |
| | chrF++ | 11.17 | 2.24 | 12.90 | **40.52** | 13.69 | 21.98 | 2.18 | 23.22 | **48.66** | 2.77 | 41.13 | **49.47** |
| **Average Score** | | 30.90 | 30.81 | 31.98 | **39.16** | 34.30 | 38.30 | 32.66 | 39.95 | **51.82** | 34.77 | 50.68 | **57.26** |

Table 2: **Base model results on Māori** (`mri`) benchmarks. Our `BYOL-mri` (4B-CPT) model obtains an average score of 47.72 outperforming the Gemma-3 (12B-PT) model, which achieves 44.88.

| Benchmarks | | 1B − 2B Models | | | | 4B − 8B Models | | | | | 12B+ Models | | |
|---|---|---|---|---|---|---|---|---|---|---|---|---|---|
| | | Llama-3.2 (1B) | Qwen-3 (1.7B-Base) | Gemma-3 (1B-PT) | BYOL-mri (1B-CPT) | Llama-3.1 (8B) | Apertus (8B-2509) | Qwen-3 (8B-Base) | Gemma-3 (4B-PT) | BYOL-mri (4B-CPT) | Qwen-3 (14B-Base) | Gemma-3 (12B-PT) | BYOL-mri (12B-CPT) |
| Global MMLU-Lite (Singh et al., 2024a) | | 26.75 | **34.25** | 22.00 | 24.25 | 38.00 | 35.25 | 42.00 | 39.00 | **45.50** | 47.00 | 45.00 | **49.00** |
| ARC-Easy (Clark et al., 2018) | | 25.63 | 26.05 | 26.35 | **30.81** | 26.60 | 25.72 | 26.81 | 26.52 | **43.73** | 26.14 | 29.50 | **41.16** |
| ARC-Hard (Clark et al., 2018) | | 18.86 | 19.62 | 18.52 | **22.87** | 20.05 | 20.90 | 21.42 | 21.08 | **32.17** | 22.35 | 23.46 | **32.59** |
| MGSM (Shi et al., 2022) | | 2.00 | **6.80** | 0.40 | 2.80 | 13.20 | 15.60 | **33.20** | 14.00 | 24.00 | 37.60 | 42.80 | **52.00** |
| XCOPA (Ponti et al., 2020) | | 52.60 | 52.80 | 52.20 | **57.00** | 55.80 | 54.80 | 51.80 | 52.80 | **60.60** | 53.60 | 55.60 | **61.20** |
| XStoryCloze (Lin et al., 2021b) | | 47.85 | 47.78 | 49.04 | **56.59** | 51.36 | 54.40 | 51.03 | 51.75 | **63.34** | 52.42 | 58.44 | **64.00** |
| PIQA (Bisk et al., 2020) | | 53.59 | 53.81 | 52.45 | **57.29** | 54.57 | 55.44 | 54.62 | 54.30 | **61.86** | 55.55 | 56.64 | **61.43** |
| HellaSwag (Zellers et al., 2019) | | 26.98 | 27.16 | 26.98 | **30.83** | 28.23 | 29.22 | 27.84 | 28.68 | **37.80** | 28.51 | 31.78 | **38.11** |
| XNLI 2.0 (Upadhyay & Upadhya, 2023) | | 33.57 | 32.38 | 32.61 | **39.36** | 35.21 | 36.63 | 35.85 | 34.97 | **44.57** | 40.98 | 41.66 | **44.51** |
| XWinograd (Tikhonov & Ryabinin, 2021) | | 49.41 | 48.66 | 49.95 | **57.33** | 52.83 | 53.05 | 52.30 | 52.83 | **59.68** | 52.19 | 56.47 | **62.67** |
| Belebele (Bandarkar et al., 2024) | | 26.22 | **30.44** | 27.44 | 27.67 | 34.78 | 36.67 | 43.33 | 34.44 | **47.78** | 48.33 | 59.11 | **63.56** |
| FLORES-200 (Costa-Jussà et al., 2022) (mri→eng) | BLEU | 2.94 | 0.39 | 3.37 | **17.45** | 16.36 | 18.93 | 8.96 | 14.35 | **25.93** | 17.31 | 23.26 | **30.15** |
| | chrF++ | 20.09 | 10.31 | 21.11 | **40.74** | 40.23 | 43.21 | 32.89 | 37.34 | **49.78** | 40.42 | 46.91 | **53.04** |
| FLORES-200 (Costa-Jussà et al., 2022) (eng→mri) | BLEU | 0.51 | 0.08 | 0.44 | **18.16** | 4.05 | 5.50 | 0.21 | 2.59 | **24.41** | 0.55 | 10.55 | **25.05** |
| | chrF++ | 13.24 | 3.42 | 14.68 | **42.69** | 25.63 | 27.48 | 5.05 | 21.14 | **49.55** | 9.03 | 36.08 | **49.68** |
| **Average Score** | | 24.98 | 25.55 | 30.29 | **37.71** | 36.65 | 37.57 | 36.78 | 36.07 | **47.72** | 39.55 | 44.88 | **51.77** |

Sec. 2.2). For Chichewa, Table 3 shows that our `BYOL-nya` (4B-M) model obtains an average performance boost of 15.20% over Apertus (8B-Instruct) model, and a 1.00% gain over a ∼7× larger model Gemma-3 (27B-IT). Similar trends can be observed for `BYOL-mri` in Table 4. Despite these strong language-specific improvements, our models also preserve English performance, as shown in Table 17.

We further evaluate the generative capability of our chat models under an LLM-as-a-judge setting. For 1,000 questions from the MultiWikiQA (Smart, 2025) reading comprehension dataset, we generate responses from all competing LLMs and compare them in a pairwise manner using GPT-5-chat as the judge. The evaluator selects the answer that is closer to the reference under a forced-choice protocol, i.e., no ties allowed. See Annex F.6 for the evaluation prompt template. The win-rate results in Fig. 4 show that our 4B models exceed the performance of substantially larger baselines. The `BYOL` (12B-M) outputs are consistently preferred by the GPT-5-chat judge across both Chichewa and Māori, surpassing Gemma-3 (27B-IT) and achieving performance on par with GPT-4o.

### 3.1.3 Ablation Studies

We perform ablation experiments on Chichewa using the 4B `BYOL` model unless mentioned otherwise.

Table 3: **Instruction-tuned model results on Chichewa** (`nya`) language benchmarks. Zero-shot evaluation on 12 datasets shows our `BYOL-nya` (M) models consistently achieve state-of-the-art performance.

| Benchmarks | | 4B – 8B Models | | | | | 12B+ Models | | | | |
|---|---|---|---|---|---|---|---|---|---|---|---|
| | | Llama-3.1 (Instruct-8B) | Apertus (8B-Inst-2509) | Qwen-3 (8B) | Gemma-3 (4B-IT) | BYOL-nya (4B-M) | Qwen-3 (14B) | GPT-OSS (20B) | Gemma-3 (12B-IT) | Gemma-3 (27B-IT) | BYOL-nya (12B-M) |
| Global MMLU-Lite (Singh et al., 2024a) | | 33.30 | 34.83 | 36.45 | 45.36 | **53.62** | 38.38 | 41.47 | 54.97 | 62.64 | **66.15** |
| ARC-Hard chat (Clark et al., 2018) | | 26.54 | 32.51 | 17.66 | 33.28 | **50.43** | 27.82 | 31.57 | 55.38 | **66.13** | 64.76 |
| MGSM (Shi et al., 2022) | | 8.80 | 2.40 | 2.40 | 11.20 | **30.00** | 3.60 | 3.50 | 37.60 | 38.80 | **40.80** |
| XCOPA (Ponti et al., 2020) | | 50.40 | 51.20 | 51.40 | 52.20 | **66.40** | 54.00 | 52.20 | 54.00 | 54.80 | **65.60** |
| XStoryCloze (Lin et al., 2021b) | | 48.97 | 50.76 | 48.78 | 49.31 | **59.23** | 51.36 | 44.87 | 55.53 | 58.31 | **62.61** |
| PIQA (Bisk et al., 2020) | | 51.41 | 52.39 | 51.25 | 52.77 | **61.75** | 52.01 | 50.44 | 55.28 | 58.38 | **63.76** |
| HellaSwag (Zellers et al., 2019) | | 29.88 | 30.58 | 29.00 | 29.10 | **45.32** | 29.54 | 25.72 | 35.08 | 40.99 | **49.16** |
| XNLI 2.0 (Upadhyay & Upadhya, 2023) | | 33.05 | 35.81 | 32.63 | 35.75 | **38.18** | 33.41 | 32.10 | 38.18 | 35.67 | **42.51** |
| XWinograd (Tikhonov & Ryabinin, 2021) | | 51.98 | 50.80 | 51.98 | 52.41 | **66.42** | 52.30 | 50.16 | 50.37 | 58.93 | **66.20** |
| Belebele (Bandarkar et al., 2024) | | 27.11 | 34.56 | 22.11 | 29.00 | **55.00** | 22.89 | 21.44 | 43.22 | 52.67 | **62.44** |
| FLORES (Costa-Jussà et al., 2022) (nya→eng) | BLEU | 8.68 | 11.53 | 6.11 | 11.97 | **24.96** | 8.38 | 1.70 | 20.08 | 22.84 | **27.13** |
| | chrF++ | 30.84 | 37.71 | 27.08 | 35.26 | **49.21** | 30.57 | 22.41 | 44.18 | 47.51 | **50.77** |
| FLORES (Costa-Jussà et al., 2022) (eng→nya) | BLEU | 1.92 | 1.48 | 0.46 | 2.80 | **13.31** | 0.94 | 1.00 | 0.98 | 8.17 | **13.65** |
| | chrF++ | 22.23 | 18.77 | 9.95 | 25.38 | **48.91** | 14.94 | 18.70 | 37.86 | 44.86 | **49.43** |
| TruthfulQA (Lin et al., 2021a) | | 30.97 | 31.70 | 30.84 | 28.76 | **36.11** | 33.05 | 36.47 | 17.38 | 29.01 | **36.96** |
| **Average Score** | | 34.27 | 35.69 | 31.66 | 36.91 | **50.89** | 34.14 | 33.16 | 44.54 | 49.90 | **55.47** |

Table 4: **Instruction-tuned model results on Māori** (`mri`) benchmarks. Zero-shot evaluation is performed.

| Benchmarks | | 4B–8B Models | | | | | 12B+ Models | | | | |
|---|---|---|---|---|---|---|---|---|---|---|---|
| | | Llama-3.1 (Instruct-8B) | Apertus (8B-Inst-2509) | Qwen-3 (8B) | Gemma-3 (4B-IT) | BYOL-mri (4B-M) | Qwen-3 (14B) | GPT-OSS (20B) | Gemma-3 (12B-IT) | Gemma-3 (27B-IT) | BYOL-mri (12B-M) |
| Global MMLU-Lite (Singh et al., 2024a) | | 30.44 | 31.05 | 33.27 | 35.10 | **47.64** | 44.68 | 34.61 | 43.52 | **54.64** | 52.48 |
| ARC-Hard chat (Clark et al., 2018) | | 31.91 | 27.30 | 34.04 | 37.50 | **51.11** | 35.75 | 32.85 | 43.09 | 49.66 | **59.22** |
| MGSM (Shi et al., 2022) | | 15.20 | 4.80 | 7.20 | 10.80 | **27.60** | 1.60 | 1.60 | 27.20 | **41.60** | 38.80 |
| XCOPA (Ponti et al., 2020) | | 54.60 | 52.00 | 52.60 | 53.60 | **57.60** | 53.60 | 53.80 | 54.40 | 52.80 | **59.80** |
| XStoryCloze (Lin et al., 2021b) | | 50.89 | 50.83 | 50.03 | 51.29 | **57.78** | 51.03 | 48.58 | 54.00 | 54.53 | **58.37** |
| PIQA (Bisk et al., 2020) | | 51.52 | 56.69 | 54.13 | 54.41 | **60.17** | 53.70 | 53.54 | 53.10 | 56.37 | **60.28** |
| HellaSwag (Zellers et al., 2019) | | 31.20 | 29.12 | 27.57 | 28.21 | **35.73** | 28.24 | 24.80 | 32.13 | 31.23 | **37.46** |
| XNLI 2.0 (Upadhyay & Upadhya, 2023) | | 35.31 | 34.13 | 34.85 | 33.81 | **42.55** | 34.89 | 34.31 | 39.50 | 37.78 | **42.51** |
| XWinograd (Tikhonov & Ryabinin, 2021) | | 49.52 | 51.12 | 50.80 | 51.55 | **56.26** | 49.95 | 50.37 | 50.16 | 51.87 | **57.33** |
| Belebele (Bandarkar et al., 2024) | | 28.00 | 33.22 | 22.89 | 25.56 | **50.67** | 22.89 | 22.00 | 47.89 | 50.33 | **62.78** |
| FLORES (Costa-Jussà et al., 2022) (mri→eng) | BLEU | 12.78 | 14.22 | 11.83 | 11.64 | **26.02** | 15.95 | 1.55 | 19.82 | 21.70 | **28.14** |
| | chrF++ | 37.28 | 40.45 | 35.83 | 34.60 | **50.75** | 40.48 | 21.31 | 43.90 | 46.79 | **52.27** |
| FLORES (Costa-Jussà et al., 2022) (eng→mri) | BLEU | 6.69 | 4.42 | 2.71 | 5.53 | **22.97** | 6.00 | 1.47 | 11.79 | 15.40 | **24.28** |
| | chrF++ | 30.75 | 26.50 | 23.38 | 28.00 | **48.48** | 29.34 | 18.87 | 36.94 | 41.51 | **49.60** |
| TruthfulQA (Lin et al., 2021a) | | 38.19 | 39.66 | 42.47 | 36.23 | **42.59** | 42.23 | 40.88 | 31.09 | 22.64 | **49.69** |
| **Average Score** | | 37.29 | 36.68 | 36.08 | 36.97 | **48.38** | 37.57 | 33.66 | 42.84 | 45.52 | **52.35** |

**Identifying the baseline LLM and MT system.** To determine the base model for adaptation, we evaluate Llama-3.1 (Grattafiori et al., 2024), Qwen-3 (Yang et al., 2025), and Gemma-3 (Team et al., 2025) on RTTBench-Mono using the round-trip translation approach (Sec. 2.1.2), where English sentences from 25 domains are translated into Chichewa and then back into English. We then compare the reconstructed English sentences with the original sentences. Table 5 shows that Gemma-3 achieves the highest BLEU, chrF++, and embedding-similarity scores, and we therefore adopt it as our baseline model.

Using the same RTT setup, we evaluate MT systems and find Azure Translator to be the best performing: Table 6 reports the fidelity scores, and Fig. 10 (in E.2) shows that it leads in 18 of the 25 domains. We use it to generate synthetic Chichewa data by translating English text into Chichewa.

**Effect of data mixture.** We evaluate four data mixtures for continual pretraining of the 4B `BYOL-nya` model: C1 uses monolingual raw Chichewa data, C2 replaces it with our refined Chichewa corpus, C3 is a refined bilingual data mixture, and C4 further includes synthetic Chichewa obtained by translating refined English text. Table 7 shows a steady improvement from C1 to C4. C4 provides the best results, increasing the Chichewa average score from 48.77 (C1) to 51.82, while preserving English performance. Per-dataset ablation results are provided in Table 18.

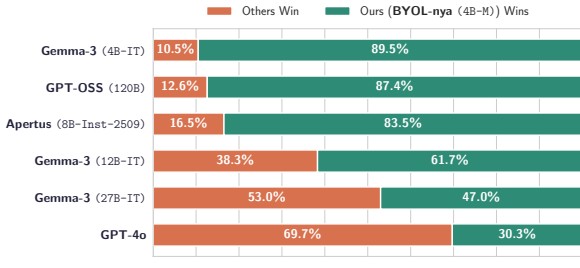

(a) `BYOL-nya` (4B-M) model performance.

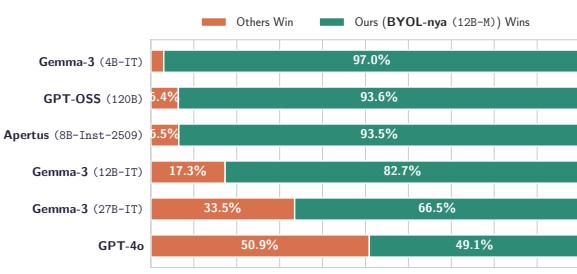

(b) `BYOL-nya` (12B-M) model performance.

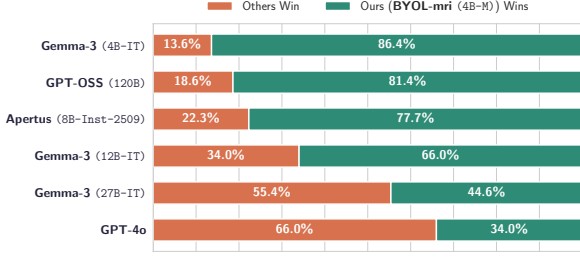

(c) `BYOL-mri` (4B-M) model performance.

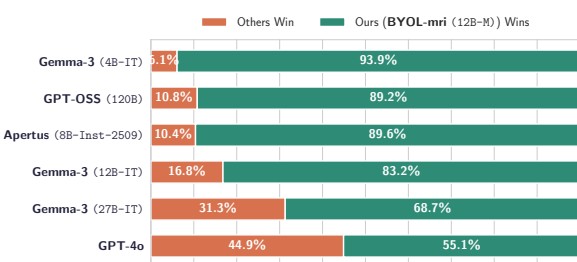

(d) `BYOL-mri` (12B-M) model performance.

Figure 4: **LLM-as-a-judge win–loss comparisons** on Multi-Wiki-QA (Smart, 2025). Pairwise evaluations against competing LLMs using GPT-5-chat as the judge. Our `BYOL` models achieve strong win rates on Chichewa and Māori; in particular, our 12B models, in (b) and (d), surpass Gemma-3 (27B-IT) and perform on par with GPT-4o.

Table 5: **Baseline LLM selection ablation** on RTTBench-Mono. Gemma-3 performs best and is therefore used for adaptation.

Table 6: **MT system selection ablation** on RTTBench-Mono. Azure Translator achieves the best overall performance and leads on 18 of 25 domains; see Fig. 10 for per-domain scores.

| | Llama-3.1 (8B-Instruct) | Qwen-3 (14B) | Gemma-3 (12B-IT) |
|---|---|---|---|
| BLEU ↑ | 3.07 | 3.33 | 11.01 |
| chrF++ ↑ | 15.86 | 20.01 | 34.21 |
| Similarity↑ | 28.66 | 20.29 | 54.34 |

| | NLLB-200 (3.3B) | MADLAD-400 (7B-MT) | GPT-4o (OpenAI) | Google (Translate) | Azure (Translator) |
|---|---|---|---|---|---|
| BLEU ↑ | 18.29 | 22.91 | 34.02 | 42.40 | 44.94 |
| chrF++ ↑ | 42.76 | 47.26 | 59.91 | 64.95 | 67.58 |
| Similarity↑ | 69.54 | 73.04 | 84.84 | 87.21 | 87.98 |

Table 7: **Data mixture ablation.** Continual pretraining of the `BYOL-nya` (4B-CPT) model under different data mixtures is performed. Average score across several datasets is reported; see Table 18 for per-dataset results. Gemma-3 (4B-PT) baseline scores are 39.95 (`nya`) and 65.17 (`eng`).

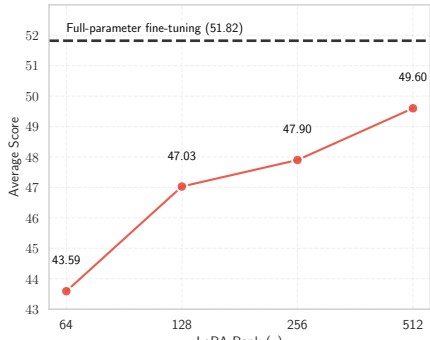

Figure 5: **LoRA vs. full-parameter** CPT of `BYOL-nya` 4B. LoRA improves with rank but remains below full-param tuning (51.82).

| Datasets | Language | C1 | C2 | C3 | C4 |
|---|---|---|---|---|---|
| FineWeb2 | nya | ✓ | ✗ | ✗ | ✗ |
| FineWeb2 → Refine (Sec. 2.2) | nya | ✗ | ✓ | ✓ | ✓ |
| FineWeb-Edu | eng | ✗ | ✗ | ✗ | ✗ |
| FineWeb-Edu → Refine (Sec. 2.2) | eng | ✗ | ✗ | ✓ | ✓ |
| FineWeb-Edu → Refine → Translate | eng→nya | ✗ | ✗ | ✗ | ✓ |
| **Average Score** | nya | 48.77 | 49.44 | 49.60 | **51.82** |
| | eng | 64.58 | 65.24 | **65.37** | 65.29 |

**LoRA vs. full-parameter finetuning.** We assess the impact on the performance of the 4B `BYOL-nya` model when updating either all parameters or only low-rank adapters. Figure 5 shows that LoRA (Hu et al.,

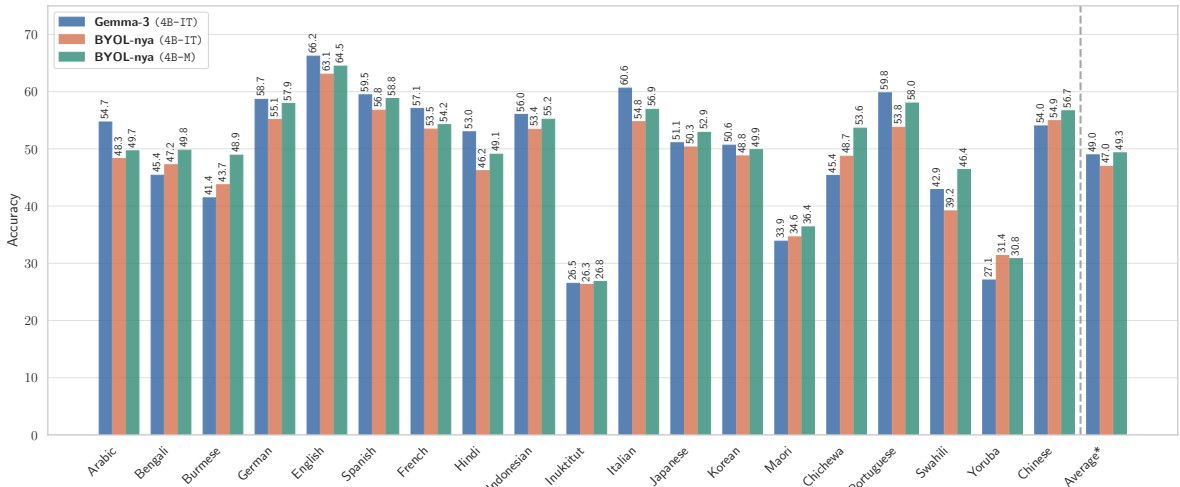

Figure 6: **Impact of model merging on multilingual performance**. Evaluation on Global MMLU-Lite dataset (Singh et al., 2024a). Average⋆ accuracy excludes English and Chichewa to measure multilingual retention.

Table 8: **Effect of model merging on bias and toxicity.** Merged models `BYOL` (M) demonstrate safety characteristics much closer to the baseline than the unmerged models `BYOL` (IT).

| Benchmarks | 4B | | | | | 12B | | | | |
|---|---|---|---|---|---|---|---|---|---|---|
| | Gemma-3 (IT) | BYOL-nya (IT) | BYOL-nya (M) | BYOL-mri (IT) | BYOL-mri (M) | Gemma-3 (IT) | BYOL-nya (IT) | BYOL-nya (M) | BYOL-mri (IT) | BYOL-mri (M) |
| BBQ (Parrish et al., 2022) ↑ ↑ | 59.34 | 48.66 | 54.25 | 45.80 | 55.39 | 67.43 | 54.37 | 70.62 | 58.21 | 69.33 |
| ToxiGen (Hartvigsen et al., 2022) ↑ | 81.49 | 42.45 | 77.55 | 43.30 | 79.47 | 86.17 | 60.64 | 86.49 | 60.11 | 86.49 |
| RealToxicity Prompts (Gehman et al., 2020) ↓ | 0.35 | 4.79 | 1.44 | 5.87 | 1.53 | 0.21 | 4.26 | 0.85 | 4.36 | 0.85 |

2022) performance improves with increasing rank, from 43.59 (r=64) to 49.60 (r=512), but remains below full-parameter tuning, which reaches 51.82. Based on these results, we adopt full-parameter training for all our models.

**Impact of model merging.** We examine the impact of model merging on multilingual performance by comparing the (1) generalist multilingual model Gemma-3 (IT), (2) language-specialist model `BYOL-nya` (IT), and (3) merged model `BYOL-nya` (M) obtained using equation 2. Figure 6 shows that the unmerged `BYOL-nya` (4B-IT) model yields strong gains on Chichewa but degrades performance on many other languages. In contrast, the merged model `BYOL-nya` (4B-M) restores the multilingual capability of the Gemma-3 (IT) model across nearly all languages while retaining the improvements on Chichewa. Overall, model merging enables language-specific specialization without sacrificing multilinguality, showing that the procedure in equation 2 effectively balances expert and generalist representations.

In addition to preserving multilingual accuracy, model merging also retains the safety characteristics of the generalist baseline. Table 8 reports bias and toxicity scores for the merged models, `BYOL-nya` (M) and `BYOL-mri` (M), compared with their unmerged IT variants, and the baseline Gemma-3 (IT). We evaluate performance on three English benchmarks: BBQ (Parrish et al., 2022) for social bias, and ToxiGen (Hartvigsen et al., 2022) and RealToxicityPrompts (Gehman et al., 2020) for toxicity. The merged models closely match baseline Gemma-3 (IT), whereas the unmerged IT models show the weakest safety performance. This indicates that merging maintains alignment and safety while adding low-resource language expertise, without any additional training or alignment steps.

**Choice of model merging weights.** Although equation 2 uses two coefficients, in the ablation we examine a one-dimensional slice by enforcing $\alpha + \beta = 1$ and reparameterizing $\alpha = 1 - \lambda$ and $\beta = \lambda$, with $\lambda \in [0, 1]$.

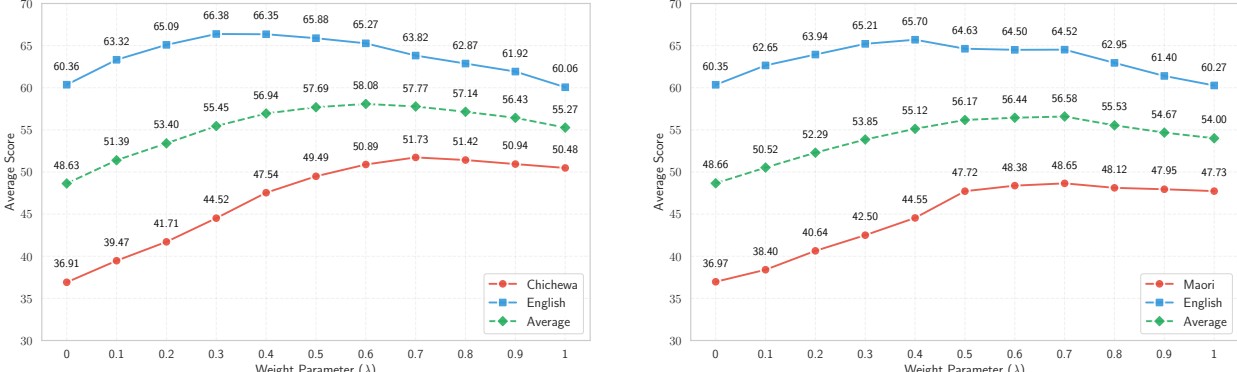

Figure 7: **Effect of model merging weight. Left:** `BYOL-nya` performance on Chichewa. **Right:** `BYOL-mri` performance on Māori. Target-language accuracy increases with larger $\lambda$, while English performance declines; the best overall trade-off occurs near $\lambda = 0.6$.

Figure 7(left) shows that increasing $\lambda$ steadily improves Chichewa performance, rising from 36.91 at $\lambda = 0$ (the pure generalist baseline) to a peak of 51.73 at $\lambda = 0.7$. English performance follows the opposite trend: it peaks at small $\lambda$ values (66.38 at $\lambda = 0.3$) and gradually declines as the expert's influence grows. The bilingual average reaches its best overall score (58.08) at $\lambda = 0.6$, which we use for all merged Chichewa models. A similar pattern appears for Māori, as shown in Fig. 7(right).

### 3.2 Translation-Mediated LLM Access for Inuktitut

#### 3.2.1 Experimental Details

**Implementation details.** We train two Transformer models (Vaswani et al., 2017): one for Inuktitut to English translation and one for the reverse direction. Each model uses 9 encoder and 9 decoder layers with an embedding size of 512, 8 attention heads, and feed-forward size of 2048 dimensions. We use the BPE (Sennrich et al., 2016b) tokenizer with a shared vocabulary of 4096 tokens for both source and target text. We train models using the Adam optimizer (Kingma & Ba, 2015) for 200K iterations with a batch size of 16K tokens. The learning rate follows the Noam schedule (Vaswani et al., 2017) with a 10K-step warmup. For regularization, we apply dropout (Srivastava et al., 2014) and label smoothing (Szegedy et al., 2016), both set to 0.1. After training, we average the last five checkpoints saved at 5K-step intervals (Edunov et al., 2018).

**Datasets.** For training the translation models, we use one publicly available dataset, the Nunavut Hansard (NH 3.0) corpus (Joanis et al., 2020), and two internal datasets originating from children's books and news articles. In addition to these human-translated datasets, we also include synthetic back-translated data collected from various sources (Tiedemann, 2012; Schwenk et al., 2021; Nguyen & Daumé III, 2019). Details on MT datasets are summarized in Appendix B.3. We convert Inuktitut text from syllabics to the romanized version (Joanis et al., 2020). We apply several filtering operations to clean the data, including removing duplicate sentence pairs, discarding sentences shorter than 3 tokens or longer than 256 tokens, and filtering out pairs with a source–target character length ratio greater than 1.3. We apply on-the-fly data augmentation (Post et al., 2023) to improve generalization, including random punctuation removal, diacritic stripping, casing variation, and a copy mechanism that replaces the source sequence with the target sequence to enable identity mapping.

#### 3.2.2 Performance Evaluation

**Evaluation of translators.** We evaluate the translation accuracy of both LLMs and NMT models in the Inuktitut ↔ English setting. Results in Table 9 and Table 10 show that our NMT models achieve state-of-the-art performance. Averaged across all datasets, our method provides a 3.64 BLEU gain over Azure Translator when translating into English, and a 4.31 BLEU gain when translating out of English. Among

Table 9: **Inuktitut → English** translation results. BLEU and chrF++: higher is better.

| | | | NH 3.0 (Dev-Test) (Joanis et al., 2020) | | NH 3.0 (Test) (Joanis et al., 2020) | | News Articles (Internal) | | Children Books (Internal) | | Average | |
|---|---|---|---|---|---|---|---|---|---|---|---|---|
| | | | BLEU | chrF++ | BLEU | chrF++ | BLEU | chrF++ | BLEU | chrF++ | BLEU | chrF++ |
| LLM | GPT-4o | 0-shot | 9.59 | 28.45 | 11.29 | 29.90 | 9.08 | 32.24 | 11.02 | 30.04 | 10.25 | 30.16 |
| | | 5-shot | 13.27 | 31.77 | 15.35 | 33.56 | 12.82 | 34.49 | 12.84 | 31.50 | 13.57 | 32.83 |
| | GPT-4.1 | 0-shot | 8.44 | 27.68 | 5.59 | 21.69 | 7.39 | 30.88 | 10.11 | 29.50 | 7.88 | 27.94 |
| | | 5-shot | 12.14 | 30.76 | 14.16 | 32.48 | 10.39 | 32.31 | 12.73 | 31.26 | 12.36 | 31.70 |
| | GPT-5-Reasoning | 0-shot | 12.80 | 33.10 | 14.44 | 34.99 | 9.88 | 34.47 | 10.79 | 31.47 | 12.00 | 33.51 |
| | | 5-shot | 16.33 | 35.75 | 18.02 | 38.17 | 12.35 | 35.94 | 12.50 | 32.81 | 14.80 | 35.67 |
| NMT | Azure Translator | – | 31.31 | 49.29 | 34.76 | 52.11 | 28.01 | 49.80 | 22.56 | 42.97 | 29.16 | 48.54 |
| | Ours | – | **35.73** | **52.44** | **40.24** | **55.89** | **28.49** | **50.06** | **26.72** | **46.48** | **32.80** | **51.22** |

Table 10: **English → Inuktitut** translation results. BLEU and chrF++: higher is better.

| | | | NH 3.0 (Dev-Test) (Joanis et al., 2020) | | NH 3.0 (Test) (Joanis et al., 2020) | | News Articles (Internal) | | Children Books (Internal) | | Average | |
|---|---|---|---|---|---|---|---|---|---|---|---|---|
| | | | BLEU | chrF++ | BLEU | chrF++ | BLEU | chrF++ | BLEU | chrF++ | BLEU | chrF++ |
| LLM | GPT-4o | 0-shot | 1.13 | 12.61 | 1.28 | 12.01 | 0.61 | 13.22 | 0.50 | 13.19 | 0.88 | 12.76 |
| | | 5-shot | 1.96 | 16.90 | 2.56 | 17.46 | 1.11 | 19.93 | 1.21 | 16.13 | 1.71 | 17.61 |
| | GPT-4.1 | 0-shot | 1.20 | 14.10 | 0.97 | 11.78 | 0.84 | 15.68 | 0.65 | 15.66 | 0.92 | 14.31 |
| | | 5-shot | 2.63 | 20.69 | 3.34 | 21.23 | 1.68 | 23.34 | 2.00 | 21.50 | 2.41 | 21.69 |
| | GPT-5-Reasoning | 0-shot | 4.43 | 23.95 | 5.68 | 24.42 | 2.45 | 24.07 | 1.91 | 25.58 | 3.62 | 24.51 |
| | | 5-shot | 6.92 | 29.62 | 6.61 | 28.31 | 3.75 | 29.20 | 4.91 | 29.69 | 5.55 | 29.21 |
| NMT | Azure Translator | – | 15.14 | 43.76 | 17.17 | 44.69 | 6.89 | 42.69 | 8.57 | 42.65 | 11.94 | 43.45 |
| | Ours | – | **18.82** | **44.59** | **21.08** | **46.34** | **12.27** | **46.57** | **14.82** | **45.29** | **16.25** | **45.70** |

Figure 8: **LLM accuracy on Global MMLU-Lite** (Singh et al., 2024a) under three evaluation settings: (1) English text input, (2) direct Inuktitut text input, and (3) a translation-mediated LLM access (Inuktitut→English→LLM). Results show a large degradation when evaluating directly in Inuktitut and a notable accuracy recovery when our machine translator is used as an intermediate step.

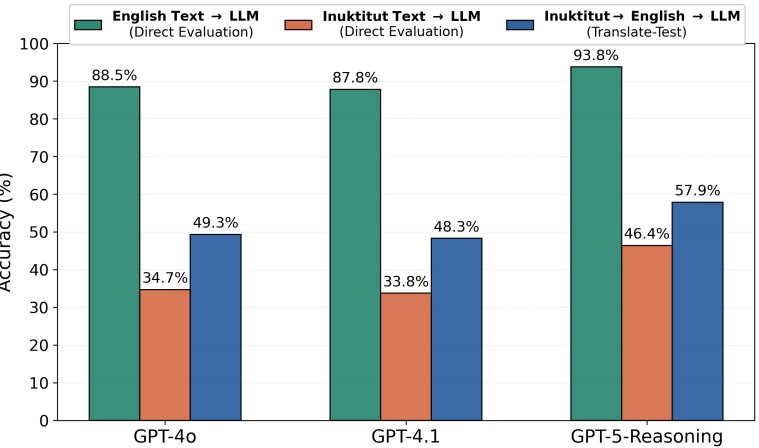

LLMs, the reasoning model (GPT-5) performs better than GPT-4o and GPT-4.1, and 5-shot prompting provides noticeable improvements over zero-shot. However, even with few-shot inference, LLM accuracy remains far below that of dedicated NMT systems.

**Evaluation of translation-mediated LLM access.** We evaluate how LLMs perform on the Global MMLU-Lite benchmark (Singh et al., 2024a) under three settings: (1) direct English text input, (2) direct Inuktitut text input, and (3) translation-mediated LLM access ((Inuktitut → English → LLM). Figure 8 shows that while all LLMs are highly accurate in English, their performance drops drastically (over 50%) when prompted directly in Inuktitut, indicating a weak understanding of the language. In contrast, inserting our MT model (Inuktitut→English) significantly recovers the lost accuracy in all LLMs. Specifically, compared to direct Inuktitut inference, translation-mediated access yields gains of 14.6% for GPT-4o, 14.42% for GPT-4.1, and 11.5% for GPT-5 (reasoning).

## 4 Limitations and Future Work

### 4.1 Limitations

**Safety evaluation scope.** Our safety analysis (Table 8) demonstrates that model merging preserves the bias and toxicity characteristics of the English-aligned baseline. However, these evaluations are conducted on English benchmarks. Cross-lingual safety leaks, including translation-based jailbreaks (Shen et al., 2024; Üstün et al., 2024), remain a concern, and dedicated target-language safety evaluation is needed before deployment in sensitive domains.

**Data scarcity.** Despite our data expansion efforts, data scarcity remains a central constraint: many LRLs still lack sufficient pretraining text, high-quality instruction and safety data, and robust language-specific evaluation sets (Romanou et al., 2024). Although we release human-translated Global MMLU-Lite benchmarks for Chichewa, Māori, and Inuktitut, broader evaluation coverage for LRLs requires sustained, community-driven data creation across modalities.

### 4.2 Future Work

**LLM safety in low-resource languages.** Future work should expand low-resource safety datasets that separately cover harmful requests, bias, and toxicity, combining careful translation of English resources with new human-curated, community-developed data that reflects local norms and realistic usage contexts (Hernández-Cano et al., 2025). A complementary direction is to build specialized safety guard models for individual low-resource languages.

**Extension to multilingual LLMs.** A natural next step is to move from single-language to multilingual specialization by integrating groups of related or typologically diverse languages (Martins et al., 2025; Dou et al., 2025), either by merging multiple language-specific experts into a shared backbone or by continual pretraining on a mixed multilingual dataset. Extending the pipeline to larger backbones (e.g., Gemma-3 27B (Team et al., 2025)) may further improve performance and clarify how benefits scale with model size.

**Scaling to more languages.** While this work demonstrates the `BYOL` pipeline on three languages, a natural next step is to apply it to additional languages, potentially in parallel. This could involve grouping typologically related languages and training a shared expert per language family, then merging it into the generalist backbone to broaden coverage efficiently. Extending the pipeline to larger backbones (e.g., Gemma-3 27B (Team et al., 2025)) may further improve performance and clarify how benefits scale with model size.

**Extension to speech.** Many LRLs are primarily spoken, making speech the most natural interface for accessing LLM-based tools. Future work should extend `BYOL` beyond text by integrating speech-to-text and text-to-speech components. Recent progress in multilingual ASR (Omnilingual et al., 2025) suggests a promising direction for building end-to-end speech–LLM stacks for low-resource settings.

## 5 Conclusion

In this paper, we proposed `BYOL`, an open framework for bringing LRLs into LLMs. `BYOL` is guided by two key inputs: (i) an initial assessment of existing AI tools to identify the most suitable open-weight LLMs and MT systems for a target language, and (ii) a language digital-resource classification that assigns each language to one of four levels, Extreme-Low, Low, Mid, and High, based on the volume of available web-scale text. The classification determines the integration strategy: direct instruction finetuning for mid- and high-resource languages, additional continual pretraining for low-resource languages, and translation-based inclusion for extreme-low-resource settings. The best tools identified through the initial assessment were then used to support data curation and to serve as baseline models for training within the selected pathway. Using `BYOL`, we instantiated the Low-resource path for Chichewa and Māori and trained two families of language-centric models, yielding four models—`BYOL-nya` (M) and `BYOL-mri` (M) at 4B and 12B parameters. Across 12 benchmarks, these models achieved an average improvement of around 12% over a strong multilingual baseline. Under LLM-as-a-judge evaluation, `BYOL-nya` (12B-M) and `BYOL-mri` (12B-M) performed on par with GPT-4o

on a question-answering benchmark, establishing a new state of the art among open models for Chichewa and Māori. For languages with extremely limited digital presence, we further explored translation-mediated inclusion on Inuktitut by training a neural machine translation system that yielded approximately +4 BLEU over a commercial baseline across three datasets, and showed that translation-mediated LLM use yields a ∼14% accuracy gain over direct inference. Finally, we released human-translated versions of Global MMLU-Lite in Chichewa, Māori, and Inuktitut, improving the reliability of multiple-choice LLM evaluation for these languages. Looking ahead, we hope `BYOL` will provide the NLP community with a practical, open recipe for extending LLM support to additional underrepresented languages and for releasing the data, models, and benchmarks needed to advance inclusive multilingual AI.

## 6 Broader Impact Statement

This work aims to expand LLM capabilities to underserved language communities, with direct implications for equitable access to AI-enabled services in education, healthcare, and economic participation. While the intended impact is positive, several potential negative impacts should be noted. First, our data refinement pipeline uses large multilingual LLMs to rephrase and expand native-language text, which may inadvertently alter authentic linguistic patterns or culturally specific content; we mitigate this by explicitly instructing the refinement model to preserve meaning and cultural intent while removing toxic or harmful material during refinement (see prompt in Appendix F.1). Nonetheless, we recommend engaging with native-speaker communities in data curation decisions, particularly for Indigenous languages where data-sovereignty considerations apply. Second, our safety analysis demonstrates that model merging preserves the bias and toxicity characteristics of the English-aligned baseline (Table 8); however, these evaluations are conducted in English, and users deploying the released models in target languages should perform additional in-language safety testing, as discussed in Sec. 4.1. Finally, because the data-curation pipeline involves LLM-generated and LLM-refined text, downstream outputs may inherit biases from the upstream models; we encourage users to apply appropriate safeguards when deploying in sensitive domains.

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

# A   Related Work

**Multilingual pretraining datasets.** Large-scale multilingual datasets (Penedo et al., 2025; Kydlíček et al., 2025; De Gibert et al., 2024; Nguyen et al., 2024; Weber et al., 2024; Laurençon et al., 2022; Wenzek et al., 2020; Conneau et al., 2020; Xue et al., 2021) are the key component of building effective multilingual LLMs. Early corpora such as CC-100 (Wenzek et al., 2020), mC4 (Xue et al., 2021), and HPLT (De Gibert et al., 2024; Burchell et al., 2025) rely on uniform, language-agnostic filtering pipelines, which often lead to uneven coverage and degraded quality for low- and mid-resource languages. FineWeb2 (Penedo et al., 2025) introduces a language-adaptive filtering and deduplication pipeline covering 1,868 language–script pairs. It provides detailed per-language metadata and yields monolingual models that outperform counterparts trained on HPLT (De Gibert et al., 2024), HPLT-2 (Burchell et al., 2025), CulturaX (Nguyen et al., 2024), and CC-100 (Wenzek et al., 2020; Conneau et al., 2020). FinePDFs (Kydlíček et al., 2025) complements this work with a 3T-token, 475M-document corpus spanning 1,733 language–script pairs, built from a dedicated PDF extraction and OCR pipeline.

In parallel, a growing number of high-quality *language-specific* pretraining datasets has emerged for individual languages, including Arabic (ArabicWeb24[7], Arabic-101B (Aloui et al., 2024)), Hindi and Telugu (Sangraha (Khan et al., 2024)), Hindi (Odaigen (Shantipriya et al., 2024)), French (Croissant (Faysse et al., 2024)), Russian (OmniaRussica[8]), Thai (SeaCommonCrawl (Dou et al., 2025)), Turkish (VNGRS-WebCorpus (Turker et al., 2024)), Chinese (Tiger-Bot[9], MAP-CC (Du et al., 2024)), Icelandic (The Icelandic Gigaword Corpus[10]), and Catalan[11]. These datasets highlight the benefits of language-aware curation for improving downstream model quality.

**Multilingual and language-centric LLMs.** Recent progress in multilingual LLMs has been driven by both closed-weight and open-weight frontier systems with broad language coverage, including mT5 (Xue et al., 2021), Aya-101 (Üstün et al., 2024), Aya-23 (Aryabumi et al., 2024), Qwen-3 (Yang et al., 2025), and Gemma-3 (Team et al., 2025). Gemma-3 notably revised its training mixture relative to Gemma-2 (Team et al., 2024) by increasing multilingual data, adding monolingual and parallel corpora, and mitigating language imbalance using a UNIMAX-inspired sampling strategy (Chung et al., 2023). Open and region-focused initiatives have expanded accessible multilingual coverage: Aya-101 increases breadth and improves instruction mixture balance (Üstün et al., 2024); Aya-23 adopts a depth-focused decoder-only design over 23 languages (Aryabumi et al., 2024); Sailor2 targets Southeast Asian languages through large-scale continual pretraining with open recipes (Dou et al., 2025); and Apertus emphasizes transparency and full reproducibility across a wide multilingual scope (Hernández-Cano et al., 2025). Europe-centric efforts such as EuroLLM and Teuken further underscore the importance of regional coverage and tokenizer design (Martins et al., 2025; Ali et al., 2024).

Complementing these multilingual generalist models, several language-centric open-weight models demonstrate the value of targeted specialization for mid- and high-resource settings, including Arabic (JAIS (Sengupta et al., 2023)), Hindi (NANDA (Choudhury et al., 2025)), French (Lucie-7B (Gouvert et al., 2025), CroissantLLM (Faysse et al., 2024)), Finnish (Poro-34B (Luukkonen et al., 2024)), Catalan (Àguila-7B (Barcelona Supercomputing Center (Projecte AINA), 2023)), Kazakh (Sherkala-Chat (Koto et al., 2025), KazLLM-1.0-8B (ISSAI, Nazarbayev University, 2024)), Greek (Llama-Krikri-8B (Roussis et al., 2025)) and Swahili (UlizaLlama (Jacaranda Health, 2023)).

Despite these advances, the long tail of low- and extreme-low-resource languages still lacks a unified, open, and tier-aware integration framework that links realistic data availability to concrete adaptation strategies. BYOL addresses this gap by introducing a four-tier digital-resource taxonomy and mapping each tier to an appropriate integration path—direct finetuning, continual pretraining, or translation-mediated inclusion.

---

[7]https://huggingface.co/blog/MayFarhat/arabicweb24

[8]https://omnia-russica.github.io/

[9]https://github.com/TigerResearch/TigerBot

[10]https://clarin.is/en/

[11]https://huggingface.co/datasets/projecte-aina/catalan_general_crawling

**Model Merging.** Model merging has emerged as an effective strategy for consolidating the complementary strengths of multiple LLMs. Early efforts demonstrated that finetuned models sharing the same pretrained backbone can be combined through simple parameter averaging, often yielding merged models that outperform the individual experts (Wortsman et al., 2022; Matena & Raffel, 2022; Gupta et al., 2020). Subsequent research explored more structured and non-linear merging techniques (Yadav et al., 2023; Yu et al., 2024), aiming to improve generalization and robustness across diverse downstream tasks. Alongside these technical advances, a growing body of work highlights the safety risks of merging: harmful or misaligned behaviors can transfer directly from one model to the merged result (Hammoud et al., 2024). This has motivated techniques that identify and manipulate safety subspaces, allowing aligned models to be fused with domain-specific experts while preserving desirable behaviors (Yu et al., 2024).

Complementary lines of work apply merging to domain-specialized experts in safety, coding, and reasoning (Wu et al., 2025; Ahmadian et al., 2024). Shadow-FT (Wu et al., 2025) adopts a paired base–instruct setup, finetuning the base model and grafting its updates onto the instruct model, and reports consistent gains over alternative merging schemes. Ahmadian et al. (2024) show that objective-driven merging of safety and general-purpose experts, including language-focused experts, outperforms simply mixing their training data, particularly in multilingual settings. Merging has also been extended to multilingual settings, enabling the construction of task-capable LLMs for high-resource languages without requiring supervised finetuning data in the target language (Tao et al., 2024). Our work builds on these developments with a goal of bringing low-resource language expertise into the baseline LLM while preserving its multilingual and safety behaviors, without requiring additional alignment training.

# B Training Datasets

## B.1 Continual Pretraining Datasets

Table 7 (Sec. 3.1.3) summarizes the four dataset configurations (C1–C4) used in our continual pretraining ablation study. The goal is to quantify, relative to a low-resource raw corpus (C1), the individual and combined contributions of refined low-resource corpora (C2), refined English corpora (C3), and their combination with the translation of refined English data (C4). The raw pretraining data for C1 are drawn from the multilingual FineWeb2 corpus (Penedo et al., 2025), while the English corpus is derived from the high-quality FineWeb-Edu English dataset (Penedo et al., 2024a).

We use `GPT-5-mini` to refine the English corpus and `GPT-5` (reasoning) to refine the target-language corpora, following the prompt structure in Annex F.1. Although many large open- and closed-weight language models could be used to refine English text, refinement in low-resource languages such as Chichewa and Māori requires particular care, as it demands strong comprehension and generation capabilities in those languages. For this reason, we use the larger-capacity `GPT-5 (reasoning)` model for the low-resource corpora, where modeling errors are more likely to propagate into downstream training stages. The English data is added in the same proportion as the low-resource language data, yielding a 1:1:1 mixture (in number of tokens) of refined low-resource text, refined English, and translated refined English into the low-resource language. This mixture in C4 dataset contains approximately 433M tokens for Chichewa and 745M tokens for Māori.

## B.2 Instruction Finetuning Datasets

Table 11 summarizes the composition of our dataset mixtures for supervised finetuning. The Aya Dataset and Aya Collection (Singh et al., 2024b) are among the few large-scale efforts that pair human-annotated multilingual instructions with scaled multilingual expansion via templating and machine translation. The Aya Dataset provides 204K human-written prompt–completion pairs across 65 languages, including Chichewa with 688 samples, while the Aya Collection broadens coverage by applying speaker-designed templates and translating selected instruction datasets into 101 languages. However, to the best of our knowledge, no large-scale, language-dedicated human-curated instruction dataset currently exists for Chichewa or Māori. To address this gap, we construct two instruction dataset mixtures specifically designed to support finetuning in these languages. Our mixtures integrate machine-translated subsets of SmolTalk2 (Bakouch et al., 2025) with additional translated data from five high-resource languages in the Aya Dataset (Singh et al., 2024b),

Table 11: **Dataset mixture used for supervised finetuning** of Chichewa and Māori LLMs.

| Dataset | | Language | # Samples | # Prompt tokens | # Response tokens | # Total tokens |
|---|---|---|---|---|---|---|
| **Chichewa SFT datasets** | | | | | | |
| Aya subset[*] (Singh et al., 2024b) | → MT | nya | 10,884 | 512,865 | 1,372,908 | 1,885,773 |
| SmolSent/Doc (Caswell et al., 2025) | | (eng, nya) | 7,992 | 242,220 | 186,844 | 429,064 |
| SmolTalk2 subset[‡] (Bakouch et al., 2025) | | eng | 350,577 | 60,325,447 | 126,499,856 | 186,825,303 |
| SmolTalk2 subset[‡] (Bakouch et al., 2025) | → MT | nya | 350,175 | 85,530,155 | 164,008,260 | 249,538,415 |
| **Total (Chichewa)** | | | 719,628 | 146,555,311 | 292,067,868 | 438,678,555 |
| **Māori SFT datasets** | | | | | | |
| Aya subset[⊗] (Singh et al., 2024b) | → MT | mri | 10,196 | 559,460 | 1,392,256 | 1,951,716 |
| Alpaca-Maori-cleaned (Upadhayay & Behzadan, 2024) | | mri | 41,601 | 1,181,880 | 9,128,222 | 10,310,102 |
| SmolTalk2 subset[‡] (Bakouch et al., 2025) | | eng | 350,577 | 60,325,447 | 126,499,856 | 186,825,303 |
| SmolTalk2 subset[‡] (Bakouch et al., 2025) | → MT | mri | 350,108 | 96,637,518 | 184,777,617 | 281,415,135 |
| **Total (Māori)** | | | 752,482 | 158,704,305 | 321,797,951 | 480,502,256 |

[*] Aya subsets from five high-resource languages, English, Italian, German, French, and Spanish, were used as sources for translation, in addition to the 688 native Chichewa samples. [⊗] Aya subsets from five high-resource languages were used as sources for translation. [‡] Samples from non-reasoning subsets of SmolTalk2 are drawn in English and machine-translated (MT) into the target language $l$.

Table 12: **Inuktitut–English paired datasets** for training and testing MT models. News Article and Children Books datasets, labeled as internal, are provided by the Government of Nunavut, Canada.

| Dataset | # Training Samples | # Dev-Test Samples | # Test Samples |
|---|---|---|---|
| Nunavut-Hansard (NH 3.0) (Joanis et al., 2020) | 1.3 million | 2,658 | 3,573 |
| News Articles (Internal) | 16,428 | – | 864 |
| Children Books (Internal) | 13,204 | – | 692 |

Table 13: **Composition of the English monolingual dataset** used for back-translation.

| Data Source Domain | Datasets Included | (Random) Samples |
|---|---|---|
| Wikipedia & General Knowledge | WikiMatrix (Schwenk et al., 2021), Simple Wikipedia (Contributors), Tatoeba (Community, 2020) | 212,500 |
| News & Commentary | News Commentary (Tiedemann, 2012), Global Voices (Nguyen & Daumé III, 2019), AG News (Zhang et al., 2015), XSum (Narayan et al., 2018) | 195,000 |
| Instructional & How-To | WikiHow (Koupaee & Wang, 2018) | 45,000 |
| Q&A and Long-Form Text | GooAQ (Khashabi et al., 2021), Natural Questions (Kwiatkowski et al., 2019), ELI5 (Fan et al., 2019), SQuAD (Rajpurkar et al., 2016) | 30,000 |
| Conversational & Talks | TED Talks (Cettolo et al., 2012) | 15,000 |
| **Total** | | **497,500** |

extended to both Chichewa and Māori. For Chichewa, we further incorporate samples from the SmolSent and SmolDoc train-paired datasets (Caswell et al., 2025), while for Māori we include cleaned data from the alpaca-maori corpus (Upadhayay & Behzadan, 2024). English instruction samples from SmolTalk2 are retained in both mixtures to ensure bilingual consistency and cross-lingual alignment. These mixtures yield SFT datasets of approximately 438M tokens (146M prompt tokens / 292M response tokens) for Chichewa and 480M tokens (158M prompt tokens / 321M response tokens) for Māori.

### B.3 Machine Translation Datasets

For training our MT models, we use three human-translated datasets, summarized in Table 12. In addition to these datasets, we also include synthetic back-translated data. For English→Inuktitut training, we back-translate 70,000 Inuktitut sentences from FineWeb2 (Penedo et al., 2025) into English. And for Inuktitut→English model training, we back-translate 497,500 English sentences from various sources, listed in Table 13.

## C Evaluation Datasets

This section describes the benchmarks used to evaluate our adapted LLMs (`BYOL-nya` and `BYOL-mri`) on Chichewa and Māori, as well as the English benchmarks used to assess their English knowledge performance. All evaluations are conducted with the LM Evaluation Harness (lm-eval) (Gao et al., 2024), a standard framework widely used in the Large Language Model (LLM) community.

Table 14:  **Benchmarks used to evaluate our base models after continual pre-training.**

| Benchmark | Languages | Category | Task | n-shot | Metric | Norm |
|---|---|---|---|---|---|---|
| **General & STEM reasoning** | | | | | | |
| Global MMLU-Lite⋆ (Singh et al., 2024a) | Chichewa, Māori | General knowledge/Reasoning | MCQ | 5 | Accuracy | |
| ARC-Easy‡ (Clark et al., 2018) | Chichewa, Māori | Science reasoning (easy) | MCQ | 0 | Accuracy | Char–Len |
| ARC-Hard‡ (Clark et al., 2018) | Chichewa, Māori | Science reasoning (hard) | MCQ | 25 | Accuracy | Char–Len |
| MGSM‡ (Shi et al., 2022) | Chichewa, Māori | Math reasoning | Generation | 8 | Accuracy (EM) | |
| BBH (Suzgun et al., 2022) | English | Complex Reasoning | Generation | 3 | Accuracy (EM) | |
| GPQA Diamond (Rein et al., 2024) | English | Graduate Science | MCQ | 5 | Accuracy | |
| **Commonsense & story** | | | | | | |
| XCOPA‡ (Ponti et al., 2020) | Chichewa, Māori | Causal reasoning | MCQ | 5 | Accuracy | |
| XStoryCloze‡ (Lin et al., 2021b) | Chichewa, Māori | Story completion/commonsense | MCQ | 5 | Accuracy | |
| PIQA‡ (Bisk et al., 2020) | Chichewa, Māori | Physical commonsense | MCQ | 0 | Accuracy | Char–Len |
| HellaSwag‡ (Zellers et al., 2019) | Chichewa, Māori | Sentence completion | MCQ | 10 | Accuracy | Char–Len |
| **NLI, coreference, and reading/QA** | | | | | | |
| XNLI 2.0‡ (Upadhyay & Upadhya, 2023) | Chichewa, Māori | Natural language inference | MCQ | 5 | Accuracy | |
| XWinograd‡ (Tikhonov & Ryabinin, 2021) | Chichewa, Māori | Coreference resolution | MCQ | 0 | Accuracy | |
| Belebele⋆ (Bandarkar et al., 2024) | Chichewa, Māori | Reading comprehension | MCQ | 1 | Accuracy | |
| **Translation** | | | | | | |
| FLORES-200⋆ (Costa-Jussà et al., 2022) | Chichewa, Māori | Translation | Generation | 1 | BLEU, chrF++ | |

⋆Human-expert translation. ‡Machine translation. EM refers to Exact Match extraction and MCQ to multiple-choice question answering.

## C.1    Base Model Benchmarking Datasets

Table 14 summarizes the benchmarks and experimental settings used to evaluate our base models after continual pretraining (`BYOL-nya`-CPT and `BYOL-mri`-CPT). We report results on fourteen datasets: twelve task-specific benchmarks in both English and the target language, and two English-only benchmarks, GPQA Diamond (Suzgun et al., 2022) and BIG-bench Hard (BBH) (Suzgun et al., 2022), which evaluate graduate-level scientific reasoning and complex problem-solving abilities. Among the target-language benchmarks, three (Global MMLU-Lite (Singh et al., 2024a), Belebele (Bandarkar et al., 2024), and FLORES-200 (Costa-Jussà et al., 2022)) were translated into Chichewa and Māori by professional human translators, while the remaining benchmarks were translated using Azure's MT system.

## C.2    Instruction-Tuned Model Benchmarking Datasets

Table 15 lists the benchmarks and experimental settings used to evaluate our chat models after instruction finetuning (`BYOL-nya`-IT, `BYOL-mri`-IT) and model merging (`BYOL-nya`-M, `BYOL-mri`-M). We evaluate on sixteen benchmarks spanning seven task categories, plus an additional question-answering benchmark, Multi-Wiki-QA (Smart, 2025), which is used only for the LLM-as-a-judge evaluation. Among the sixteen benchmarks, four are English-only and cover general scientific reasoning (GPQA Diamond (Suzgun et al., 2022), BBH (Suzgun et al., 2022)), instruction-following (IFEval (Zhou et al., 2023a)), and code generation (HumanEval (Chen et al., 2021)). The remaining benchmarks are evaluated in both English and the two target languages, Chichewa and Māori. We also perform a multilingual ablation on Global MMLU-Lite (Singh et al., 2024a) across 19 languages.

## C.3    LLM as Judge Evaluation

Following prior work on *LLM-as-a-Judge* evaluation (Üstün et al., 2024; Rafailov et al., 2023; Dubois et al., 2023), we design a structured prompt template for pairwise preference assessment in Chichewa and Māori. We instantiate this protocol on a question-answering task using 1,000 randomly selected samples from the Multi-Wiki-QA dataset (Smart, 2025). In each comparison, the judge model receives the context, question, a reference answer, and two candidate responses presented in randomized order. It is instructed to (i) produce a comparative rationale evaluating linguistic correctness, instruction adherence, factual accuracy, semantic comprehension, and grammatical fluency in the target language; (ii) select a preferred response label from {A, B}; and (iii) assign individual quality scores (0–5) to each candidate according to these criteria. The

Table 15: **Benchmarks used to evaluate our instruct-tuned and merged models.** The chat template is enabled for all benchmarks.

| Benchmark | Languages | Category | Task | n-shot | Metric |
|---|---|---|---|---|---|
| **General & STEM reasoning** | | | | | |
| Global MMLU-Lite⋆ (Singh et al., 2024a) | 19 languages[1] | General knowledge/Reasoning | MCQ (Gen.) | 0 | Accuracy |
| ARC Challenge chat[‡] (Clark et al., 2018) | Chichewa, Māori | Science reasoning (hard) | MCQ (Gen.) | 0 | Accuracy (EM) |
| MGSM[‡] (Shi et al., 2022) | Chichewa, Māori | Math reasoning | Generation | 0 | Accuracy (EM) |
| BBH (Suzgun et al., 2022) | English | Complex reasoning | Generation | 0 | Accuracy (EM) |
| GPQA Diamond (Rein et al., 2024) | English | Graduate Science | MCQ (Gen.) | 0 | Accuracy (EM) |
| **Commonsense & story** | | | | | |
| XCOPA[‡] (Ponti et al., 2020) | Chichewa, Māori | Causal reasoning | MCQ (LH) | 0 | Accuracy |
| XStoryCloze[‡] (Lin et al., 2021b) | Chichewa, Māori | Story completion/commonsense | MCQ (LH) | 0 | Accuracy |
| PIQA[‡] (Bisk et al., 2020) | Chichewa, Māori | Physical commonsense | MCQ (LH) | 0 | Accuracy |
| HellaSwag[‡] (Zellers et al., 2019) | Chichewa, Māori | Sentence completion | MCQ (LH) | 0 | Accuracy |
| **Reading comprehension & QA** | | | | | |
| XNLI 2.0[‡] (Upadhyay & Upadhya, 2023) | Chichewa, Māori | Natural language inference | MCQ (LH) | 0 | Accuracy |
| XWinograd[‡] (Tikhonov & Ryabinin, 2021) | Chichewa, Māori | Coreference resolution | MCQ (LH) | 0 | Accuracy |
| Belebele⋆ (Bandarkar et al., 2024) | Māori, Chichewa | Reading comprehension | MCQ (LH) | 0 | Accuracy |
| Multi-Wiki-QA[⊗] (Smart, 2025) | Chichewa, Māori | Reading comprehension/QA | Generation | 0 | LLM-as-judge[2] win–loss |
| **Instruction following** | | | | | |
| IFEval[3] (Zhou et al., 2023a) | English | Instruction following | Generation | 0 | Accuracy |
| **Truthfulness** | | | | | |
| TruthfulQA[‡] (Lin et al., 2021a) | Chichewa, Māori | Truthfulness | MCQ (Gen) | 0 | BLEU Accuracy |
| **Code generation** | | | | | |
| HumanEval (Chen et al., 2021) | English | Code generation | Generation | 0 | Pass@1 |
| **Translation** | | | | | |
| FLORES-200⋆ (Costa-Jussà et al., 2022) | Chichewa, Māori | Translation | Generation | 0 | BLEU, chrF++ |

⋆ Human-expert translation. ‡ Machine translation. ⊗ LLM-generated questions from Wikipedia documents.
EM refers to Exact Match extraction, MCQ to multiple-choice question answering, and LH to likelihood-based scoring.
[1] Global-MMLU-Lite (Singh et al., 2024a) originally includes 16 languages; in this paper we extend it to 19 by adding three human-expert-translated languages: Chichewa, Māori, and Inuktitut. [2] The GPT-5-chat model is used as the judge for win–loss comparisons. [3] prompt_level_loose_acc score is reported.

Table 16: **Base model results on English** benchmarks. After continual pretraining of Gemma-3 baseline, our `BYOL` models preserve English language performance. Results on Chichewa (`nya`) and Māori (`mri`) benchmarks are provided in Tables 1 and 2.

| Benchmarks | 1B | | | 4B | | | 12B | | |
|---|---|---|---|---|---|---|---|---|---|
| | Gemma-3 (PT) | BYOL-nya (CPT) | BYOL-mri (CPT) | Gemma-3 (PT) | BYOL-nya (CPT) | BYOL-mri (CPT) | Gemma-3 (PT) | BYOL-nya (CPT) | BYOL-mri (CPT) |
| Global MMLU-Lite (Singh et al., 2024a) | 25.00 | 23.50 | 23.25 | 66.75 | 65.50 | 64.25 | 76.50 | 76.75 | 76.25 |
| ARC-Easy (Clark et al., 2018) | 71.97 | 68.31 | 68.22 | 81.78 | 77.61 | 75.93 | 87.79 | 86.45 | 85.69 |
| ARC-Hard (Clark et al., 2018) | 39.51 | 40.36 | 39.25 | 58.36 | 58.28 | 59.64 | 67.83 | 68.17 | 68.00 |
| MGSM (Shi et al., 2022) | 3.60 | 1.20 | 2.80 | 46.40 | 52.40 | 50.40 | 76.80 | 80.80 | 82.00 |
| BBH (Suzgun et al., 2022) | 28.35 | 26.74 | 27.83 | 38.46 | 37.00 | 37.58 | 52.39 | 51.08 | 52.39 |
| GPQA Diamond (Rein et al., 2024) | 22.73 | 24.75 | 23.74 | 30.30 | 36.36 | 32.32 | 35.35 | 32.32 | 35.86 |
| XCOPA (Ponti et al., 2020) | 81.00 | 78.00 | 79.00 | 88.00 | 89.00 | 86.00 | 93.00 | 94.00 | 94.00 |
| XStoryCloze (Lin et al., 2021b) | 72.20 | 71.87 | 72.20 | 80.08 | 80.87 | 81.07 | 84.18 | 83.59 | 83.98 |
| PIQA (Bisk et al., 2020) | 75.03 | 73.39 | 73.56 | 79.71 | 79.05 | 78.94 | 81.83 | 81.72 | 81.34 |
| HellaSwag (Zellers et al., 2019) | 62.93 | 62.06 | 62.39 | 77.71 | 77.73 | 77.81 | 84.11 | 83.21 | 83.43 |
| XNLI 2.0 (Upadhyay & Upadhya, 2023) | 48.31 | 48.76 | 48.39 | 51.00 | 49.88 | 51.45 | 53.98 | 52.45 | 53.61 |
| Winograd (Tikhonov & Ryabinin, 2021) | 58.80 | 58.33 | 58.25 | 69.46 | 67.72 | 68.11 | 75.22 | 74.03 | 74.35 |
| Belebele (Bandarkar et al., 2024) | 27.11 | 27.33 | 28.00 | 79.22 | 77.33 | 75.44 | 92.22 | 92.22 | 92.67 |
| **Average Score** | 47.43 | 46.51 | 46.68 | 65.17 | 65.29 | 64.53 | 73.94 | 73.60 | 74.12 |

full prompt template used for this evaluation is provided in Annex F.6. We compare our models against the following competing LLMs: Gemma-3 (4B/12B/27B-IT) (Team et al., 2025), GPT-OSS (120B) (Agarwal et al., 2025), Apertus (8B-Instruct-2509) (Hernández-Cano et al., 2025), and GPT-4o. GPT-5-chat is used exclusively as the evaluation judge and is not included as a competing model.

Table 17: **Instruction-Tuned model results on English** benchmarks. Results on Chichewa (`nya`) and Māori (`mri`) benchmarks are provided in Tables 3 and 4.

| Benchmarks | 4B | | | 12B | | |
|---|---|---|---|---|---|---|
| | Gemma-3 (IT) | BYOL-nya (M) | BYOL-mri (M) | Gemma-3 (IT) | BYOL-nya (M) | BYOL-mri (M) |
| Global MMLU-Lite (Singh et al., 2024a) | 66.21 | 64.47 | 66.30 | 75.15 | 81.71 | 81.00 |
| ARC-Hard (Clark et al., 2018) chat | 75.85 | 75.77 | 76.11 | 90.10 | 88.99 | 88.82 |
| MGSM (Shi et al., 2022) | 47.20 | 57.60 | 60.00 | 58.00 | 75.20 | 76.00 |
| BBH (Suzgun et al., 2022) | 50.44 | 44.16 | 44.40 | 56.27 | 57.58 | 57.40 |
| GPQA Diamond (Rein et al., 2024) | 33.84 | 34.85 | 26.26 | 36.36 | 37.37 | 37.37 |
| XCOPA (Ponti et al., 2020) | 86.00 | 89.00 | 88.00 | 88.00 | 95.00 | 95.00 |
| XStoryCloze (Lin et al., 2021b) | 65.45 | 72.47 | 73.40 | 69.09 | 77.37 | 76.70 |
| PIQA (Bisk et al., 2020) | 69.91 | 78.13 | 78.56 | 71.16 | 81.83 | 81.45 |
| HellaSwag (Zellers et al., 2019) | 47.47 | 71.85 | 70.90 | 55.70 | 77.32 | 77.17 |
| XNLI 2.0 (Upadhyay & Upadhya, 2023) | 43.41 | 54.42 | 53.61 | 50.36 | 56.06 | 54.38 |
| XWinograd (Tikhonov & Ryabinin, 2021) | 61.25 | 67.56 | 67.72 | 65.98 | 74.35 | 74.59 |
| Belebele (Bandarkar et al., 2024) | 69.67 | 87.11 | 83.22 | 91.33 | 92.67 | 92.56 |
| truthfulqa-multi gen (Lin et al., 2021a) | 42.96 | 47.12 | 49.33 | 49.45 | 49.94 | 46.14 |
| IFEval (Zhou et al., 2023a) | 76.89 | 79.11 | 75.97 | 75.23 | 85.77 | 86.51 |
| HumanEval (Chen et al., 2021) | 68.90 | 55.49 | 53.66 | 80.04 | 78.66 | 79.27 |
| **Average Score** | 60.36 | 65.27 | 64.50 | 67.87 | 73.99 | 73.62 |

## D  Performance of BYOL Models on English

Tables 16 and 17 show that our LRL-specific `BYOL` models maintain English performance after adaptation and remain comparable to the Gemma-3 baseline.

## E  Ablation Experiment Results

### E.1  RTTBench-Mono Validation

The prompt tempelate to construct RTTBench-Mono dataset is provided in F.5. To reduce semantic drift between overlapping categories (e.g., Health vs. Beauty & Fitness), each prompt includes the domain definition and a list of confusable categories to avoid during data generation We perform dataset quality verification using two complementary approaches. First, we apply NVIDIA's domain classifier on RTTBench-Mono, which achieves 97.8% Top-1 accuracy, confirming strong alignment between each sentence and its intended domain. Second, we embed all sentences using `text-embedding-3-large` model and visualize them via t-SNE. Figure 9 shows that clusters from all 25 domains are well separated, indicating that RTTBench-Mono maintains distinct domains and avoids significant cross-domain overlap.

### E.2  Evaluation of MTs on RTT Benchmarks

Figure 10 presents a per-domain MT ablation on the RTTBench-Mono dataset using our round-trip translation setup (Sec. 2.1.2), in which English sentences from 25 domains are translated into Chichewa and back to English, and the reconstructed sentences are compared to the originals. The left panel reports per-domain sacreBLEU scores, while the right panel reports per-domain embedding-similarity scores. Across both metrics, Azure Translator achieves the highest macro-average performance and the largest number of domain wins.

### E.3  Impact of CPT Data Mixture

Table 18 reports impact of various CPT data mixtures on the performance of `BYOL-nya`.

### E.4  Ablation of LoRA vs. Full-Parameter

Table 19 shows results on the LoRA vs. full-parameter continual pretraining for `BYOL-nya` (CPT-4B).

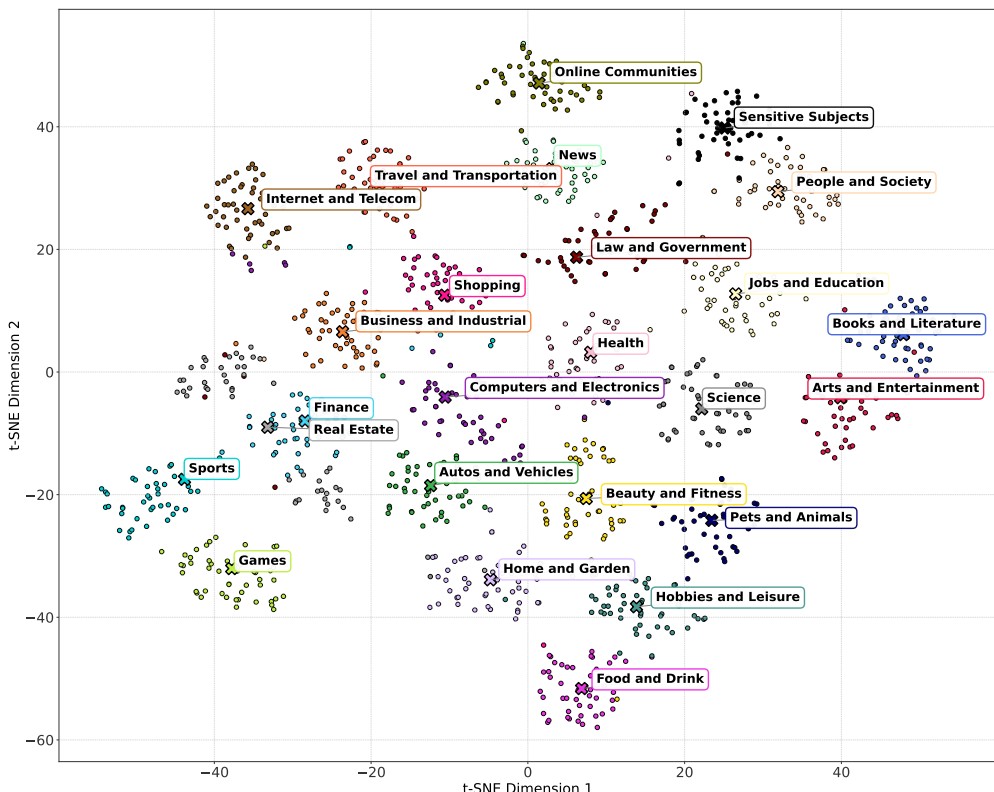

Figure 9: **t-SNE visualization of RTTBench-Mono embeddings.** Sentences from all 25 domains form well-separated clusters, indicating strong domain fidelity and minimal cross-domain overlap.

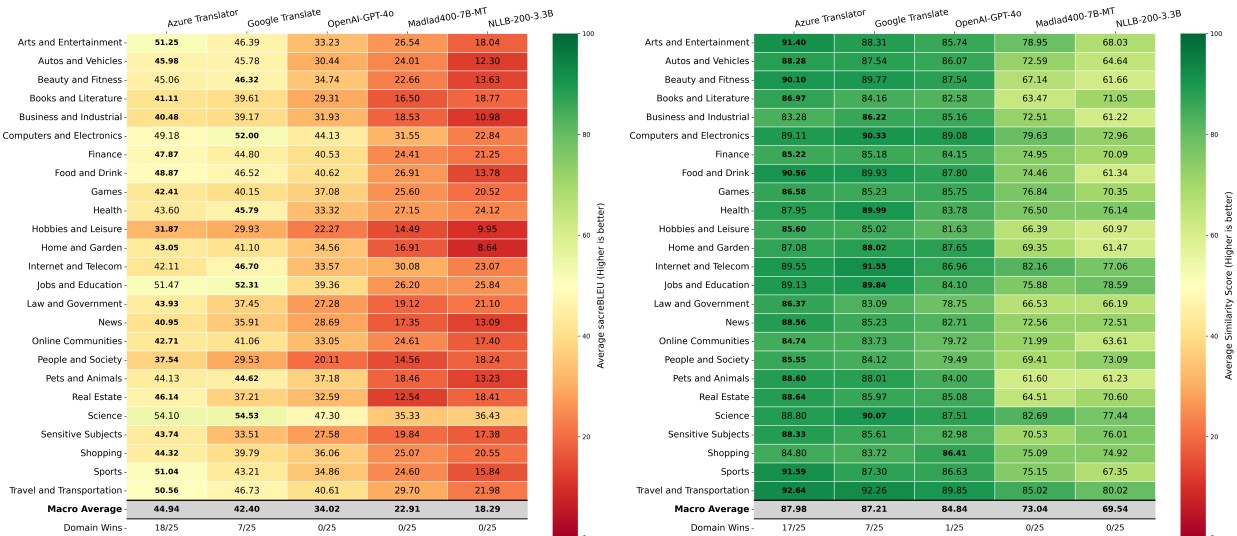

Figure 10: **Ablation of per-domain MT evaluation on RTTBench-Mono dataset.** We use the round-trip translation setup (Sec. 2.1.2), where English sentences from 25 domains are translated into Chichewa and back to English, and the reconstructed sentences are compared to the originals. **Left:** per-domain sacreBLEU scores. **Right:** per-domain embedding-similarity scores. Azure Translator achieves the highest macro-average and the most domain wins.

Table 18: **Data mixture ablation.** Continual pretraining of the `BYOL-nya` (CPT-4B) model under different data mixtures. For the definitions of the C1–C4 mixtures, see Table 7. Gemma-3 is the baseline.

| Model | | Gemma-3 (4B-PT) | | C1 | | C2 | | C3 | | C4 | |
|---|---|---|---|---|---|---|---|---|---|---|---|
| | | eng | nya | eng | nya | eng | nya | eng | nya | eng | nya |
| Global MMLU-Lite (Singh et al., 2024a) | | 66.75 | 50.75 | 64.00 | 54.00 | 63.25 | 52.50 | 65.75 | 52.50 | 65.50 | 55.25 |
| ARC-Easy (Clark et al., 2018) | | 81.78 | 30.22 | 80.68 | 42.26 | 79.76 | 41.96 | 79.53 | 42.47 | 77.61 | 48.48 |
| ARC-Hard (Clark et al., 2018) | | 58.36 | 27.13 | 59.04 | 36.35 | 59.81 | 37.63 | 59.90 | 37.88 | 58.28 | 40.61 |
| MGSM (Shi et al., 2022) | | 46.40 | 17.20 | 45.20 | 26.80 | 50.40 | 26.80 | 52.40 | 26.00 | 52.40 | 31.60 |
| BBH (Suzgun et al., 2022) | | 38.46 | - | 37.03 | - | 37.49 | - | 37.29 | - | 37.00 | - |
| GPQA Diamond (Rein et al., 2024) | | 30.30 | - | 30.30 | - | 32.32 | - | 32.32 | - | 36.36 | - |
| XCOPA (Ponti et al., 2020) | | 88.00 | 57.20 | 88.00 | 64.40 | 89.00 | 65.60 | 89.00 | 68.00 | 89.00 | 70.00 |
| XStoryCloze (Lin et al., 2021b) | | 80.08 | 54.40 | 80.61 | 63.73 | 80.68 | 64.66 | 80.81 | 64.73 | 80.87 | 65.98 |
| PIQA (Bisk et al., 2020) | | 79.71 | 54.57 | 80.41 | 61.15 | 79.54 | 63.33 | 79.11 | 62.35 | 79.05 | 63.71 |
| HellaSwag (Zellers et al., 2019) | | 77.71 | 33.45 | 78.73 | 43.61 | 78.09 | 44.80 | 77.71 | 45.59 | 77.73 | 47.31 |
| XNLI 2.0 (Upadhyay & Upadhya, 2023) | | 51.00 | 37.82 | 51.61 | 39.58 | 51.53 | 40.92 | 51.16 | 41.10 | 49.88 | 40.32 |
| XWinograd (Tikhonov & Ryabinin, 2021) | | 69.46 | 54.76 | 68.27 | 66.10 | 69.53 | 68.02 | 67.96 | 66.10 | 67.72 | 68.34 |
| Belebele (Bandarkar et al., 2024) | | 79.22 | 38.22 | 75.67 | 43.67 | 76.67 | 42.44 | 76.89 | 43.67 | 77.33 | 45.44 |
| FLORES (Costa-Jussà et al., 2022) (nya→eng) | BLEU | - | 17.28 | - | 23.22 | - | 22.82 | - | 22.93 | - | 23.87 |
| | chrF++ | - | 40.37 | - | 47.28 | - | 46.78 | - | 47.10 | - | 47.95 |
| FLORES (Costa-Jussà et al., 2022) (eng→nya) | BLEU | - | 2.24 | - | 11.42 | - | 11.72 | - | 11.86 | - | 12.79 |
| | chrF++ | - | 23.22 | - | 45.11 | - | 47.29 | - | 47.32 | - | 48.66 |
| **Average Score** | | 65.17 | 39.95 | 64.58 | 48.77 | 65.24 | 49.44 | 65.37 | 49.60 | 65.29 | 51.82 |

Table 19: **Ablation of LoRA vs. full-parameter** continual pre-training of `BYOL-nya` (CPT-4B). The accuracy of LoRA model improves with increasing rank but remains below full-parameter tuning (51.82). Gemma-3 is the baseline.

| Model | | Gemma-3 (4B-PT) | | 64 | | 128 | | 256 | | 512 | | Full-Parameter Tuning | |
|---|---|---|---|---|---|---|---|---|---|---|---|---|---|
| | | eng | nya | eng | nya | eng | nya | eng | nya | eng | nya | eng | nya |
| Global MMLU-Lite (Singh et al., 2024a) | | 66.75 | 50.75 | 64.25 | 51.25 | 65.25 | 55.25 | 64.50 | 55.25 | 64.75 | 57.00 | 65.50 | 55.25 |
| ARC-Easy (Clark et al., 2018) | | 81.78 | 30.22 | 79.08 | 34.43 | 76.47 | 38.89 | 76.68 | 39.98 | 76.18 | 43.69 | 77.61 | 48.48 |
| ARC-Hard (Clark et al., 2018) | | 58.36 | 27.13 | 59.39 | 31.91 | 58.53 | 33.53 | 58.19 | 34.39 | 57.85 | 36.09 | 58.28 | 40.61 |
| MGSM (Shi et al., 2022) | | 46.40 | 17.20 | 52.00 | 22.40 | 50.40 | 24.80 | 51.60 | 28.40 | 53.60 | 27.60 | 52.40 | 31.60 |
| BBH (Suzgun et al., 2022) | | 38.46 | - | 40.45 | - | 40.44 | - | 39.47 | - | 38.06 | - | 37.00 | - |
| GPQA Diamond (Rein et al., 2024) | | 30.30 | - | 27.78 | - | 33.84 | - | 31.31 | - | 30.81 | - | 36.36 | - |
| XCOPA (Ponti et al., 2020) | | 88.00 | 57.20 | 88.00 | 60.00 | 87.00 | 64.40 | 87.00 | 64.80 | 87.00 | 66.20 | 89.00 | 70.00 |
| XStoryCloze (Lin et al., 2021b) | | 80.08 | 54.40 | 81.14 | 58.17 | 80.21 | 60.03 | 79.88 | 60.09 | 80.41 | 63.47 | 80.87 | 65.98 |
| PIQA (Bisk et al., 2020) | | 79.71 | 54.57 | 80.09 | 55.82 | 79.65 | 59.68 | 79.65 | 59.30 | 79.22 | 61.21 | 79.05 | 63.71 |
| HellaSwag (Zellers et al., 2019) | | 77.71 | 33.45 | 77.16 | 36.32 | 76.40 | 40.92 | 76.26 | 41.46 | 76.31 | 43.79 | 77.73 | 47.31 |
| XNLI 2.0 (Upadhyay & Upadhya, 2023) | | 51.00 | 37.82 | 53.33 | 38.42 | 51.61 | 38.96 | 51.53 | 39.78 | 50.60 | 40.60 | 49.88 | 40.32 |
| XWinograd (Tikhonov & Ryabinin, 2021) | | 69.46 | 54.76 | 69.46 | 60.64 | 69.93 | 65.24 | 70.17 | 66.63 | 69.69 | 68.02 | 67.72 | 68.34 |
| Belebele (Bandarkar et al., 2024) | | 79.22 | 38.22 | 79.22 | 38.11 | 79.89 | 40.44 | 80.67 | 42.11 | 80.00 | 42.78 | 77.33 | 45.44 |
| FLORES (Costa-Jussà et al., 2022) (nya→eng) | BLEU | - | 17.28 | - | 19.76 | - | 21.90 | - | 22.60 | - | 24.22 | - | 23.87 |
| | chrF++ | - | 40.37 | - | 43.22 | - | 45.91 | - | 46.80 | - | 48.03 | - | 47.95 |
| FLORES (Costa-Jussà et al., 2022) (eng→nya) | BLEU | - | 2.24 | - | 6.51 | - | 10.13 | - | 10.22 | - | 11.33 | - | 12.79 |
| | chrF++ | - | 23.22 | - | 35.99 | - | 43.39 | - | 43.74 | - | 46.32 | - | 48.66 |
| **Average Score** | | 65.17 | 39.95 | 65.49 | 43.59 | 65.36 | 47.03 | 65.15 | 47.90 | 64.96 | 49.60 | 65.29 | 51.82 |

## E.5  Impact of Model Merging Weight

Tables 20 and 21 present detailed results on the impact of the weight-merging parameter $\lambda$ across 12 Chichewa benchmarks and 15 English benchmarks, respectively.

Table 20: **Effect of model-merging weight $\lambda$ evaluated on Chichewa benchmarks.** $\lambda = 0$ corresponds to the baseline model Gemma-3 (4B-IT), and $\lambda = 1$ to our instruction-tuned Chichewa model `BYOL-nya` (4B-IT).

| Model | | Merging weight parameter ($\lambda$) | | | | | | | | | | |
|---|---|---|---|---|---|---|---|---|---|---|---|---|
| | | 0 | 0.1 | 0.2 | 0.3 | 0.4 | 0.5 | 0.6 | 0.7 | 0.8 | 0.9 | 1 |
| Global MMLU-Lite (Singh et al., 2024a) | | 45.36 | 48.86 | 47.73 | 50.96 | 53.43 | 53.79 | 53.62 | 54.53 | 51.46 | 50.66 | 48.72 |
| ARC-Hard chat (Clark et al., 2018) | | 33.28 | 35.67 | 39.25 | 43.69 | 46.08 | 48.21 | 50.43 | 51.45 | 51.62 | 51.19 | 49.57 |
| MGSM (Shi et al., 2022) | | 11.20 | 22.00 | 21.20 | 27.20 | 28.80 | 26.40 | 30.00 | 30.00 | 27.20 | 19.60 | 16.00 |
| XCOPA (Ponti et al., 2020) | | 52.20 | 50.60 | 52.00 | 55.40 | 59.40 | 65.40 | 66.40 | 65.80 | 67.60 | 68.80 | 70.80 |
| XStoryCloze (Lin et al., 2021b) | | 49.31 | 51.56 | 53.47 | 54.60 | 56.12 | 57.71 | 59.23 | 60.82 | 60.62 | 60.42 | 60.03 |
| PIQA (Bisk et al., 2020) | | 52.77 | 52.29 | 54.30 | 56.69 | 58.54 | 61.10 | 61.75 | 62.68 | 62.73 | 63.11 | 63.22 |
| HellaSwag (Zellers et al., 2019) | | 29.10 | 31.25 | 34.15 | 37.29 | 40.52 | 43.30 | 45.32 | 47.08 | 47.54 | 47.62 | 47.21 |
| XNLI 2.0 (Upadhyay & Upadhya, 2023) | | 35.75 | 36.27 | 37.50 | 37.43 | 38.30 | 38.64 | 39.18 | 40.40 | 40.14 | 40.12 | 39.92 |
| XWinograd (Tikhonov & Ryabinin, 2021) | | 52.41 | 51.44 | 54.33 | 56.47 | 60.64 | 61.82 | 66.42 | 68.98 | 68.77 | 70.16 | 71.02 |
| Belebele (Bandarkar et al., 2024) | | 29.00 | 32.78 | 37.22 | 43.11 | 49.78 | 52.11 | 55.00 | 54.22 | 53.56 | 53.33 | 53.33 |
| FLORES (Costa-Jussà et al., 2022) (nya→eng) | BLEU | 11.97 | 14.95 | 18.36 | 21.08 | 22.80 | 24.32 | 24.96 | 25.33 | 25.49 | 25.49 | 24.63 |
| | chrF++ | 35.26 | 39.09 | 42.46 | 45.03 | 47.02 | 48.46 | 49.21 | 49.86 | 49.94 | 49.81 | 48.93 |
| FLORES (Costa-Jussà et al., 2022) (eng→nya) | BLEU | 2.80 | 4.51 | 6.49 | 9.03 | 10.68 | 12.39 | 13.31 | 13.83 | 13.99 | 14.21 | 13.87 |
| | chrF++ | 25.38 | 31.05 | 36.51 | 41.76 | 45.14 | 47.62 | 48.91 | 49.70 | 49.89 | 50.33 | 50.02 |
| TruthfulQA (Lin et al., 2021a) | | 28.76 | 30.23 | 32.07 | 29.13 | 34.27 | 38.80 | 36.11 | 36.96 | 37.33 | 37.09 | 37.45 |
| **Average Score** | | 36.91 | 39.47 | 41.71 | 44.52 | 47.54 | 49.49 | 50.89 | 51.73 | 51.42 | 50.94 | 50.48 |

Table 21: **Effect of model-merging weight $\lambda$ evaluated on English benchmarks.** $\lambda = 0$ corresponds to the baseline model Gemma-3 (4B-IT), and $\lambda = 1$ to our instruction-tuned Chichewa model `BYOL-nya` (4B-IT).

| Model | Merging weight parameter ($\lambda$) | | | | | | | | | | |
|---|---|---|---|---|---|---|---|---|---|---|---|
| | 0 | 0.1 | 0.2 | 0.3 | 0.4 | 0.5 | 0.6 | 0.7 | 0.8 | 0.9 | 1 |
| Global MMLU-Lite (Singh et al., 2024a) | 66.21 | 68.23 | 68.58 | 66.75 | 66.80 | 65.43 | 64.47 | 64.55 | 63.60 | 63.45 | 63.06 |
| ARC-Hard chat (Clark et al., 2018) | 75.85 | 76.88 | 77.56 | 77.99 | 77.99 | 75.68 | 75.77 | 73.46 | 71.42 | 68.94 | 66.21 |
| MGSM (Shi et al., 2022) | 47.20 | 57.60 | 64.80 | 68.00 | 63.20 | 60.80 | 57.60 | 53.20 | 49.60 | 48.00 | 38.80 |
| BBH (Suzgun et al., 2022) | 50.44 | 51.39 | 50.88 | 49.58 | 48.27 | 45.43 | 44.16 | 42.87 | 42.10 | 40.64 | 38.70 |
| GPQA Diamond (Rein et al., 2024) | 33.84 | 35.35 | 34.34 | 33.84 | 33.84 | 35.86 | 34.85 | 32.32 | 32.32 | 34.34 | 34.85 |
| XCOPA (Ponti et al., 2020) | 86.00 | 89.00 | 89.00 | 90.00 | 89.00 | 89.00 | 89.00 | 89.00 | 89.00 | 89.00 | 88.00 |
| XStoryCloze (Lin et al., 2021b) | 65.45 | 67.31 | 69.03 | 70.42 | 71.54 | 71.94 | 72.47 | 73.20 | 72.60 | 71.94 | 70.62 |
| PIQA (Bisk et al., 2020) | 69.91 | 72.14 | 73.56 | 75.35 | 76.93 | 77.42 | 78.13 | 79.11 | 78.78 | 79.05 | 78.45 |
| HellaSwag (Zellers et al., 2019) | 47.47 | 51.89 | 57.20 | 62.80 | 67.39 | 69.99 | 71.85 | 72.73 | 72.87 | 72.44 | 71.90 |
| XNLI 2.0 (Upadhyay & Upadhya, 2023) | 43.41 | 47.47 | 49.24 | 50.92 | 52.61 | 53.86 | 54.42 | 53.53 | 53.13 | 52.25 | 51.29 |
| XWinograd (Tikhonov & Ryabinin, 2021) | 61.25 | 61.88 | 65.35 | 67.40 | 66.93 | 67.09 | 67.56 | 67.32 | 68.82 | 68.67 | 67.64 |
| Belebele (Bandarkar et al., 2024) | 69.67 | 77.78 | 84.11 | 87.11 | 87.22 | 86.89 | 87.11 | 86.00 | 84.89 | 84.11 | 81.67 |
| TruthfulQA (Lin et al., 2021a) | 42.96 | 43.08 | 42.72 | 46.63 | 46.27 | 48.23 | 47.12 | 43.70 | 43.94 | 43.45 | 41.74 |
| IFEval (Zhou et al., 2023a) | 76.89 | 81.52 | 83.55 | 83.73 | 82.62 | 80.78 | 79.11 | 72.64 | 68.76 | 66.73 | 64.70 |
| Humaneval (Chen et al., 2021) | 68.90 | 68.29 | 66.46 | 65.24 | 64.63 | 59.76 | 55.49 | 53.66 | 51.22 | 45.73 | 43.29 |
| **Average Score** | 60.36 | 63.32 | 65.09 | 66.38 | 66.35 | 65.88 | 65.27 | 63.82 | 62.87 | 61.92 | 60.06 |

# F Prompts

## F.1 Prompt for Text Refinement

```
You are an expert content editor and enhancer.  You will receive text primarily in
{LANGUAGE_NAME} (ISO639-3 code:  {LANGUAGE_CODE}).
Your task:
    • If the text is in {LANGUAGE_NAME} or mixed with another language (e.g.,
      English), keep and refine it.
    • If the text is not in {LANGUAGE_NAME} at all, remove it completely.
For each input item:
```

- Rewrite the text for clarity, flow, and readability while preserving meaning.

- Fix grammar, spelling, and punctuation.

- Reorganize ideas logically to improve coherence.

- Replace repetitive or awkward wording with smoother alternatives, but do **not** shorten the overall text.

- Enrich the text with on-topic elaboration, nuance, or re-expression so the final output is equal to or longer than the input.

- Target length: 100-140% of original. **Absolute rule: never shorter than the input.**

- Default tone: clear, engaging, and respectful.

- Do not add unrelated facts or change intent.

- Do not alter or rewrite any direct quotations from religious scriptures (e.g., Bible, Qur'an, Hadith, Torah, Vedas, etc.). Preserve them exactly as written.

Rules:

- Expansion is required: if you reduce wording in one place, expand in another place on the same topic.

- Keep all additions factually aligned with the input.

- Respect sensitive language; maintain a respectful tone.

- Remove toxic or unsafe text.

## F.2 Prompt for Sentence Alignment in Extreme-Low-Resource Languages

You are a bilingual text alignment assistant. Given two ordered lists of sentences (Set A = source, Set B = target):

1. Align them monotonically, contiguous, non-overlapping.

2. Fix formatting issues (merge broken lines, normalize spacing/punctuation, ensure each aligned pair is a clean sentence or short paragraph).

3. Output ONLY valid JSON Lines (JSONL) format, one alignment per line:

{"source": ["<cleaned source text>"], "target": ["<cleaned target text>"]}

Rules:

- Prefer 1-1 sentence alignment; allow 1-N, M-1, or M-N if needed.

- Copy text exactly after cleaning, no translation.

- Skip pairs where source and target are in the same language or where text overlap is greater than ~70%. Do not output these at all.

- Return ONLY the JSONL output, no other text or explanations.

### F.3 Prompt for LLM-Based Enhancement of English Text for Back-Translation

```
You are an expert English copy editor preparing sentences for machine translator
training.
For each input item:

    • If the text is not natural-language English (e.g., another language, code,
      LaTeX, complex math) → skip.

    • If the text is meaningless → skip.

    • Fix spelling, grammar, agreement, casing, whitespace, and hyphenation.  Keep
      named entities intact.

    • You may freely rephrase to improve clarity and flow, staying close to the main
      theme and content.

    • If text exceeds 30 words → rewrite and summarize within 30 words.

    • Maintain sentence case and end with proper punctuation.

    • Produce clean, easy-to-understand English sentences suitable for translation.
```

### F.4 Prompt for LLM-Based Post-Editing of MT Outputs

```
You are a translation post-editor.  The {source_text} is in {source_lang}, and the
{translated_text} is in {target_lang}.
The translation may contain minor errors or wrong word choices.  Your task is to
correct only those errors with minimal edits.
Do not change sentence structure, phrasing, or style unnecessarily - preserve the
original translation as much as possible.
Do not try to make factual corrections, only fix language issues.
Output only the corrected {target_lang} sentence without any quotes, explanations, or
additional formatting.
```

### F.5 Prompt to Generate RTTBench-Mono

```
You are a domain-aware writing assistant generating English sentences that will later
be used to benchmark machine-translation systems.  Your task is to generate sentences
that are diverse in style and complexity, natural, and highly specific to the {domain}
domain.  Each sentence must be self-contained and unambiguously belong to this domain.

Domain Context and Disambiguation

    • Target Domain:  {domain}

    • Domain Description:  {domain_description}

    • Confusable Domains to Avoid:  {', '.join(confusable_domains)}

Instructions

    1. Generate exactly {count} independent sentences that are clearly and
       specifically about {domain}.
```

2. Each sentence must be unambiguously about {domain} and should feel out of place or less relevant in these related domains: {', '.join(confusable_domains)}. This is the most important rule.

3. Return the sentences as a plain numbered list (1., 2., etc.). No extra commentary.

4. Every sentence must be unique, self-contained, safe, and obviously about {domain}.

5. **Strictly enforce sentence-length quotas.** Your final output of {count} sentences **must** be composed of an exact number of sentences from each length band, as specified below:

   - Short (SENTENCE_LENGTH_BANDS[0] words): exactly {quotas[0]} sentences.
   - Medium (SENTENCE_LENGTH_BANDS[1] words): exactly {quotas[1]} sentences.
   - Long (SENTENCE_LENGTH_BANDS[2] words): exactly {quotas[2]} sentences.

   You must verify your own output to ensure this distribution is perfectly met.

6. **Vary the tone, register, and complexity.** The set of {count} sentences should include everyday/accessible style, informal with some domain-specific jargon, formal/literary, and technical/professional language. Do *not* follow a fixed pattern; the mix should look organic.

7. Cover different linguistic dimensions across your sentences:

   - sentence type: declarative, interrogative, imperative, exclamatory, factual, reasoning-based, comparative, causal, hypothetical, counterfactual, indirect speech;
   - voice: active and passive;
   - tense/aspect: past, present, future, perfect, conditional;
   - terminology: mix common domain vocabulary with more specialized jargon or acronyms;
   - entities and numerics: names, dates, currencies, units, measurements;
   - figurative language where appropriate;
   - occasional ambiguity (e.g., 'bank', 'rock');
   - co-reference and pronouns (he, she, they, it) - but not in every sentence.

8. **Final formatting rules:**

   - If a sentence needs quotation marks, use single quotes (').
   - Do not add external commentary or restate these instructions in the output.

## F.6 Prompt Template for LLM-as-a-Judge Evaluation

```
System Preamble
You are an expert evaluator of language-model outputs in {LANGUAGE_NAME}. Your task
is to compare two candidate answers to the same question in {LANGUAGE_NAME} and select
the better one.
```

```
Prompt Template
Evaluation criteria:
1)  Linguistic correctness:  the answer must be primarily in
    [LANGUAGE_NAME] and use appropriate vocabulary and style.
2)  Instruction following:  the answer should fully address
    the question and follow all given instructions.
3)  Factual accuracy and semantic comprehension with respect
    to the context and reference answer.
4)  Grammar and fluency in {LANGUAGE_NAME}.

Context:    {CONTEXT}
Question:   {QUESTION}
Reference   {ANSWER}
answer:

Answer (A):  {COMPLETION A}
Answer (B):  {COMPLETION B}

Evaluate both answers according to the criteria above and provide:
1)  A comparison in English explaining which answer is better and why.
2)  Your preference:  Answer (A) or Answer (B).
3)  A rating for each answer on a 0-5 scale based on the
    evaluation criteria (0 = unusable, 5 = excellent).

Format your response exactly as follows (no additional text):

Comparison:  <your comparison>
Preferred:  <"Answer (A)" or "Answer (B)">
Rating output A: [0-5]
Rating output B: [0-5]
```

