# OpenReview forum: "BYOL: Bring Your Own Language Into LLMs"
_TMLR — Under review for TMLR_

### Review · Reviewer_ZWvm · 2026-06-27

**Summary Of Contributions:**

The global distribution of digital text is severely skewed: of the ~7,000 living languages, fewer than ~100 have enough web presence to meaningfully shape LLM training, and ~90% of Common Crawl text comes from roughly twenty languages. This produces a self-reinforcing "AI diffusion divide" in which speakers of low-resource languages (LRLs) get degraded model quality, higher tokenization/inference costs, weaker safety behavior, and exclusion from AI-enabled services. The paper proposes **BYOL (Bring Your Own Language)**, a resource-adaptive, full-stack recipe for extending LLM capabilities to LRLs and extreme-low-resource languages (ELRLs), and instantiates it end-to-end on three typologically diverse, genuinely underserved languages: Chichewa (`nya`, Bantu/Malawi), Māori (`mri`, Polynesian), and Inuktitut (`iku`, Inuit/Canada).

The contributions are:

1. **A four-tier language resource classification.** Using per-language word counts from FineWeb2 (Penedo et al., 2025), the paper bins languages into Extreme-Low, Low, Mid, and High, and pairs corpus size with speaker population (Fig. 3). Each tier is mapped to an integration pathway: direct finetuning (mid/high), continual pretraining (low), and translation-mediated access (extreme-low).
2. **A low-resource pathway: CPT → SFT → model merging.** For low-resource languages, the paper (a) curates a CPT corpus mixing refined real FineWeb2 text, MT-translated FineWeb-Edu (synthetic LRL), and refined English at a 1:1:1 token ratio (~433M tokens for Chichewa, ~745M for Māori), with all text refined/expanded by a large LLM; (b) builds a bilingual SFT mixture from Aya, SmolTalk2, and MT-translated instructions; and (c) merges the language expert back into the generalist in weight space (Eq. 2, task-arithmetic style) to recover multilingual ability and safety. Applied to Gemma-3 (1B/4B/12B), this yields BYOL-`nya` and BYOL-`mri`.
3. **An extreme-low-resource pathway: a dedicated NMT system + Translate-Test.** For Inuktitut, the paper trains forward/back Transformer NMT models (Nunavut Hansard + internal news/children's books + back-translated FineWeb2), with LLM-based sentence alignment and LLM post-editing (GPT-5), and uses them as a translation interface to an English-centric LLM (Translate-Test, Artetxe et al., 2023).

**Strength**
- Chichewa (Bantu), Māori (Polynesian, revitalized Indigenous), and Inuktitut (Inuit, syllabics, polysynthetic) span very different morphologies and resource situations. Demonstrating the *extreme*-low-resource pathway on Inuktitut — where direct adaptation is infeasible — is more than most LRL papers attempt, and the translate-test recovery numbers (Fig. 8) are a clean, interpretable result.
- Professionally human-translated Global MMLU-Lite in three underserved languages is a tangible contribution that outlives the paper, and human-translated benchmarks for these languages are scarce. The released codebase and the RTTBench-Mono domain-balanced RTT set lower the cost of follow-up work.

**Weakness**
- No variance, seeds, or significance testing anywhere — yet the headline claims are small-margin "small beats large" comparisons. Every table is single-run point estimates. The flagship claims rest on margins that are plausibly within run-to-run noise: BYOL-`nya` 4B-CPT beats Gemma-3 12B-PT by 1.14 points (51.82 vs. 50.68; the text says "1.24," a discrepancy in itself), and BYOL-`nya` 4B-M beats Gemma-3 27B-IT by 0.99 points (50.89 vs. 49.90). A paper whose abstract and Figure 1 are built around "the 4B variants outperform the ~7× larger Gemma-3 (27B-IT)" cannot support that claim without at least multiple seeds and confidence intervals (or bootstrap CIs on the per-benchmark scores).
- Train/test translationese confound: most benchmarks are machine-translated with the same MT system used to build the training data. Per Table 14, only *3 of the target-language benchmarks are human-translated* (Global MMLU-Lite, Belebele, FLORES-200); the rest are *machine-translated* (per the ‡ footnote). Meanwhile, a third of the CPT corpus is FineWeb-Edu *translated into the target language by the best available MT system (Azure)*, and the SFT mixture is heavily MT-translated. A model trained on MT-translationese will match the distribution of MT-translated benchmarks better than the baseline, independent of genuine language ability. This confound plausibly inflates the average, and the paper never separates "human-translated benchmark" performance from "machine-translated benchmark" performance in its headline.

**Audience:**

Yes

**Audience Explanation:**

Clearly, multilingual and low-resource NLP is an active and growing TMLR-relevant area, and this paper speaks to several overlapping communities.

**Claims And Evidence:**

Yes

**Claims Explanation:**

The competitive headline claims, i.e.,"4B beats 12B-PT," "4B-M beats 27B-IT," "on par with GPT-4o", are not adequately supported in their current form. They rest on single-run point estimates with sub-1-to-1-point margins, an aggregate that mixes incommensurable metrics, a benchmark suite dominated by machine translation that shares the training data's MT system, and, for the GPT-4o comparison, an unvalidated single LLM judge operating in languages it may not understand well. The improvement over the same-size Gemma-3 baseline is real and substantial and survives on the human-translated subset, so the core "targeted adaptation helps these languages a lot" message holds; but the *magnitudes* and the *small-beats-large* framing are over-claimed relative to the statistics provided.

**Requested Changes:**

1. Add variance and significance to the headline comparisons. Provide multiple seeds (or at minimum bootstrap CIs over per-instance benchmark scores) for the small-margin claims, especially "4B beats Gemma-3 12B-PT" (+1.14) and "4B-M beats Gemma-3 27B-IT" (+0.99).
2. Validate the LLM-as-a-judge protocol (Fig. 4) in the target languages, or downgrade the "on par with GPT-4o" claim. Provide judge–human agreement on a sample of Chichewa/Māori comparisons, add a position-swap control, and ideally use ≥2 independent judges.
3. Audit the LLM-refined CPT data with native speakers. Report grammaticality/faithfulness/hallucination rates on a sampled subset of the refined Chichewa/Māori corpus, and add an ablation that isolates the contribution of refinement from added translationese. Reconcile the mandatory-expansion prompt (F.1) with the Broader Impact concern about altering authentic linguistic patterns.

---

### Review · Reviewer_xcj8 · 2026-06-29

**Summary Of Contributions:**

This paper proposes a framework, named BYOL (Bring Your Own Language), that aims to systematically and efficiently extend the capabilities of LLMs to low-resource and extremely low-resource languages.  For low-resource languages, the authors integrate data curation, continual pretraining, supervised fine-tuning, and model merging into a unified end-to-end pipeline. They validate the pipeline on Chichewa and Māori, achieving substantial improvements over baselines. For extremely low-resource languages, the paper uses Inuktitut as a case study and shows that, with the assistance of a specialized MT system, the proposed approach can also improve system accuracy.

**Strengths** : The proposed framework is conceptually sound, and the experimental results demonstrate its effectiveness. The experimental design and evaluation are also diverse.

**Weaknesses** : The evaluation lacks human validation. The framework is evaluated on only three languages, which may not be sufficient.

**Audience:**

Yes

**Audience Explanation:**

LLMs still require improvement in low-resource and extremely low-resource language settings, and this topic is likely to attract interest within the TMLR community.

**Claims And Evidence:**

No

**Claims Explanation:**

Most of the authors' claims are supported by the experimental results and analyses. However, the evaluation is conducted primarily in English and lacks human validation. The proposed system appears to be effective, but the reported performance gains may require further verification, for example, among the 12 benchmarks used by the authors, only 3 are human-translated. In addition, evaluating the framework on only three languages may be insufficient.

**Requested Changes:**

1. Have the authors considered validating the results with native speakers, or is this difficult to implement in practice?

2. Have the authors considered extending their experiments to a larger number of languages?

3. The authors mainly compare their approach with general-purpose LLMs, but there are also fine-tuned models for specific low-resource languages. Could the authors consider including comparisons with such specialized models, potentially requiring experiments on additional languages?